# The splicing factor RBM25 controls MYC activity in acute myeloid leukemia

Ying Ge[1,2,3], Mikkel Bruhn Schuster[1,2,3], Sachin Pundhir[1,2,3,4], Nicolas Rapin[1,2,3,4], Frederik Otzen Bagger[1,2,3,4], Nikos Sidiropoulos[1], Nadia Hashem[1,2,3] & Bo Torben Porse [1,2,3]

Cancer sequencing studies have implicated regulators of pre-mRNA splicing as important disease determinants in acute myeloid leukemia (AML), but the underlying mechanisms have remained elusive. We hypothesized that "non-mutated" splicing regulators may also play a role in AML biology and therefore conducted an in vivo shRNA screen in a mouse model of *CEBPA* mutant AML. This has led to the identification of the splicing regulator RBM25 as a novel tumor suppressor. In multiple human leukemic cell lines, knockdown of *RBM25* promotes proliferation and decreases apoptosis. Mechanistically, we show that RBM25 controls the splicing of key genes, including those encoding the apoptotic regulator BCL-X and the MYC inhibitor BIN1. This mechanism is also operative in human AML patients where low *RBM25* levels are associated with high MYC activity and poor outcome. Thus, we demonstrate that RBM25 acts as a regulator of MYC activity and sensitizes cells to increased MYC levels.

[1] The Finsen Laboratory, Rigshospitalet, Faculty of Health and Medical Sciences, University of Copenhagen, Ole Maaløes Vej 5, 2200 Copenhagen N, Denmark. [2] Biotech Research and Innovation Centre, University of Copenhagen, Ole Maaløes Vej 5, 2200 Copenhagen N, Denmark. [3] Novo Nordisk Foundation Center for Stem Cell Biology, DanStem, Faculty of Health Sciences, Faculty of Health and Medical Sciences, University of Copenhagen, Ole Maaløes Vej 5, 2200 Copenhagen N, Denmark. [4] The Bioinformatics Centre, Department of Biology, Faculty of Natural Sciences, University of Copenhagen, Ole Maaløes Vej 5, 2200 Copenhagen N, Denmark. These authors contributed equally: Ying Ge, Mikkel Bruhn Schuster. Correspondence and requests for materials should be addressed to B.T.P. (email: bo.porse@finsenlab.dk)

Acute myeloid leukemia (AML) is an aggressive hematological disorder for which there is an unmet medical need for novel treatment strategies. AML constitutes an arrested state of development in which leukemic blasts, resembling normal myeloid progenitor cells, fail to terminally differentiate and consequently accumulate in the bone marrow (BM) and peripheral organs. In addition, seminal work has demonstrated that AML is maintained by relatively rare populations of leukemic stem cells (LSCs) with self-renewal capacity[1,2]. Thus insights into how these cells are controlled hold the potential of serving as a starting point for the rational development of novel treatment strategies.

Recent cancer genome sequencing studies have revealed the genetics of many cancers including AML. In addition to genes encoding epigenetic regulators, transcription factors, and growth regulators, splicing factor genes are often mutated in human AML[3]. Recurrently mutated splicing factors in AML include SRSF2, SF3B1, and U2AF1 and these lesions are found in approximately 10% of patients[4,5]. The latter factors are involved in pre-mRNA splicing, a process catalyzed by the spliceosome—a major ribonucleoprotein complex that acts in a sequential manner to remove introns[6]. In addition to core spliceosome components, splicing is also influenced by a set of regulatory factors that promote or repress defined steps during the process in a pre-mRNA-specific manner resulting in a range of so-called alternatively spliced transcripts[7,8]. These transcripts may have an impact on downstream protein production via different means. Commonly, alternative splicing affects transcript stability, i.e. leads to changes in protein levels, but may also affect coding potential leading to the expression of proteins with distinct functional properties.

Despite the fact that splicing factor mutations are commonly found in AML and other hematological malignances, including myelodysplastic syndrome (MDS), it has remained largely elusive how they mediate or sustain oncogenic transformation[9]. Generally, mutations of these factors influence the splicing patterns of hundreds of pre-mRNAs and whether this malignant phenotype is driven by individual variants (and if so which) or the sum of changes has proven difficult to resolve[10]. Moreover, the finding that splicing patterns are also affected in AML patient samples with no obvious mutations in splicing-related genes suggests that splicing regulators may be affected by other means including epigenetically induced de-regulation[11].

Loss-of-function (LOF) screens using siRNA, shRNA, or CRISPR-based approaches have been used extensively for the identification of oncogenes and tumor suppressors. LOF screens are generally performed in vitro, not least due to the excellent library coverage that can be obtained in a controlled experimental setting with nearly unlimited amounts of cells. However, in vitro screens may miss genes that are important only in an in vivo setting or detect genes that are important only in an in vitro setting. Hence ideally, LOF screens should be performed in vivo in relevant model systems[12]. CEBPA is an important myeloid transcription factor that is frequently mutated in human AML, and biallelic CEBPA mutant AML constitutes a specific AML subtype[3,13,14]. The underlying genetic lesions in biallelic CEBPA mutant AML converge at the selective expression of the p30 isoform, i.e. an N-terminally truncated version of CEBPA. In contrast to full-length (p42) CEBPA, this variant is unable to facilitate E2F-mediated cell cycle repression and it also targets a slightly different set of promoters and enhancers[15–17] (manuscript in preparation). In mice, the exclusive expression of the p30 isoform from the endogenous Cebpa locus leads to the development of AML within a year and this mouse line (Lp30) therefore constitutes an excellent model for human biallelic CEBPA mutant AML[18].

Here we have conducted an in vivo shRNA screen in the context of the Cebpa mutant AML mouse model aimed at identifying novel tumor suppressors and oncogenes among potential splicing regulators. This led to the identification of RBM25, a relatively uncharacterized RNA binding protein, as a potential tumor suppressor. Using a variety of human leukemic cell lines and a hierarchical organized AML cell culture system, we demonstrated that loss of RBM25 leads to a decrease of apoptosis and an increase in the proliferation of not only bulk leukemic cells, but also cells with LSC properties. Molecularly, we showed that RBM25 controls the splicing of a number of key pre-mRNAs including those encoding the apoptotic regulator BCL-X and the MYC inhibitor, BIN1. Thus our findings provide not only novel mechanistic insights into the role of RBM25 in AML but also a proof-of-concept for the functional importance of splicing de-regulation in AML biology in general.

## Results

**Identification of novel splicing regulators in murine AML.** In order to identify splicing factors that functionally affect AML biology, we conducted an in vivo pooled shRNA screen in the Lp30 AML model. The screen was based on the assumption that when AML cells are transduced with a mixture of shRNAs, continuous proliferation of the cells will lead to the gradual depletion of shRNAs targeting tumor-promoting genes and vice versa for shRNAs targeting tumor suppressive genes.

To investigate the specific effects of factors involved in posttranscriptional pre-mRNA processing, we constructed a retroviral shRNA library containing 613 shRNAs targeting 230 known or putative splicing factor-encoding genes. The targets include all major known and putative splicing associated gene families, such as Serine/Arginine-rich proteins (SR proteins), heterogeneous nuclear ribonucleoproteins (hnRNPs), and RNA binding motif proteins (Fig. 1a and Supplementary Data 1). The library was used to transduce AML cells derived from the Lp30 mice and these cells were subsequently transplanted into secondary recipients. Following the development of secondary AML, the shRNA repertoires of input and secondary AML cells were determined by next-generation sequencing in order to quantify changes in the representation of individual shRNAs (Fig. 1b). Since we were mainly interested in targets specifically affecting growth of leukemic cells, but not of their normal hematopoietic counterparts, we also performed an in vitro counter screen in normal c-Kit-enriched murine hematopoietic progenitors (Supplementary Fig. 1a). We selected an shRNA pool size of 150 based on a preliminary screen with increasing pool sizes, followed by assessment of correlation between the replicates (Supplementary Fig. 1b).

The in vivo AML screen was performed in duplicate and we observed a near-complete coverage of the shRNA pools (Supplementary Fig. 1c) as well as moderate correlation ($R = 0.53$) between the biological replicates when comparing the fold-change (FC) in shRNA representation between start and endpoint cells (Supplementary Fig. 1d). shRNAs were ranked based on their mean FC (Fig. 1c), and a gene was scored as a hit if targeted by multiple shRNAs ranked within the 20th percentile of the most enriched or 25th percentile of the most depleted shRNAs in Lp30 cells, but not in normal c-Kit+ cells. This resulted in the identification of six enriched and eight depleted candidate genes (Supplementary Tables 1 and 2). We further excluded hits, for which the scoring shRNAs displayed poor knockdown (KD) efficiency and selected four enriched hits (Rbm25, Sfrs12ip1, Hnrnph1, and Prpf40a) and five depleted hits (Cpsf1, Cpsf6, Hnrnpab, Rbpms2, and Xab2) (Fig. 1c) for in vivo validation. To this end, we carried out in vivo competitive bone marrow

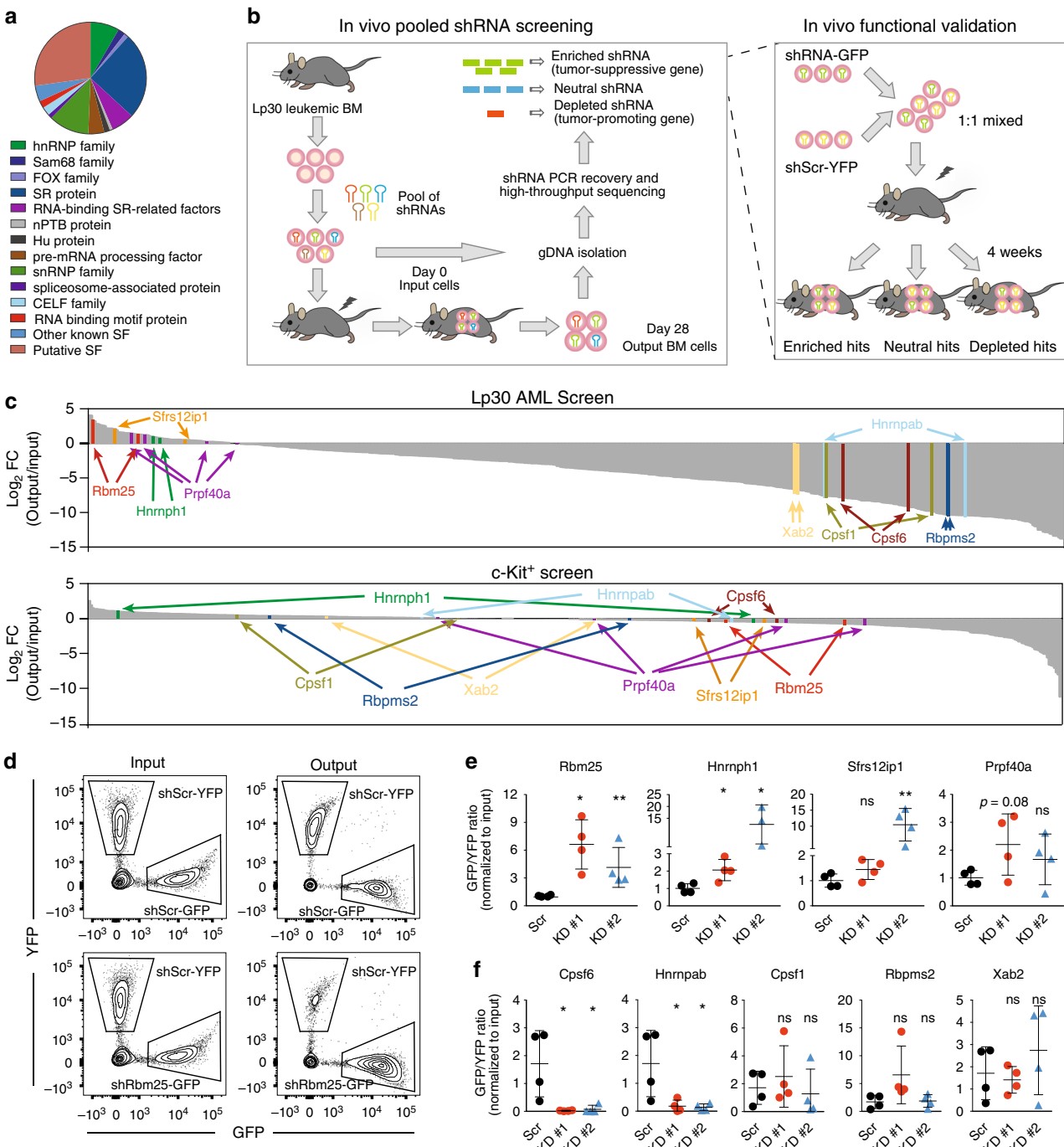

**Fig. 1** Pooled in vivo shRNA screen for essential splicing factors in the Lp30 AML model. **a** Relative proportions of splicing factor categories represented in the shRNA library. **b** Schematic outline of the in vivo shRNA screen (left panel) and the in vivo GFP/YFP competitive assay used to functionally validate candidates (right panel). In the screening procedure, sublethally irradiated recipients were transplanted with library-transduced Lp30 AML cells and allowed to develop AML, after which the relative representation of the individual shRNA inserts in output vs. input cells was assessed. For validation, cells transduced with candidate-specific shRNA/GFP and scrambled shRNA/YFP-expressing retrovirus were mixed and injected into recipients who developed AML, after which GFP/YFP ratios in output vs. input cells were quantified by flow cytometry. **c** Results of screens carried out in Lp30 AML cells (upper panel) and normal c-Kit+ cells (lower panel). The $\log_2$ fold changes in the representation of individual shRNAs in output vs. input samples are indicated. shRNAs targeting selected candidate genes are highlighted. **d** Representative example of GFP+ and YFP+ cells in input vs. output AML cells from the in vivo validation of *Rbm25*. **e, f** In vivo validation of candidate shRNAs, which were either enriched (**e**) or depleted (**f**) in the Lp30 screen (**c**). Data represent the mean ± s.d. ($n = 4$ for each group). Data were subjected to an unpaired *t* test, and asterisks indicate the following: *$P < 0.05$; **$P < 0.01$. Panel (**e**) shows representative results of two independent experiments for *Rbm25*. All replicates in **e** and **f** are biological replicates

transplantations (BMT) using Lp30 donor cells transduced with target-specific shRNA constructs coexpressing GFP along with YFP-expressing nontargeting (scrambled) shRNA-transduced competitors (Fig. 1d, see KD efficiencies in Supplementary Fig. 1e). In this assay, KD of *Rbm25* and *Hnrnph1* gave rise to significant enrichment of cells transduced with either of the two target-specific shRNAs over their scrambled shRNA-transduced counterparts (Fig. 1e), thus recapitulating the phenotype from the

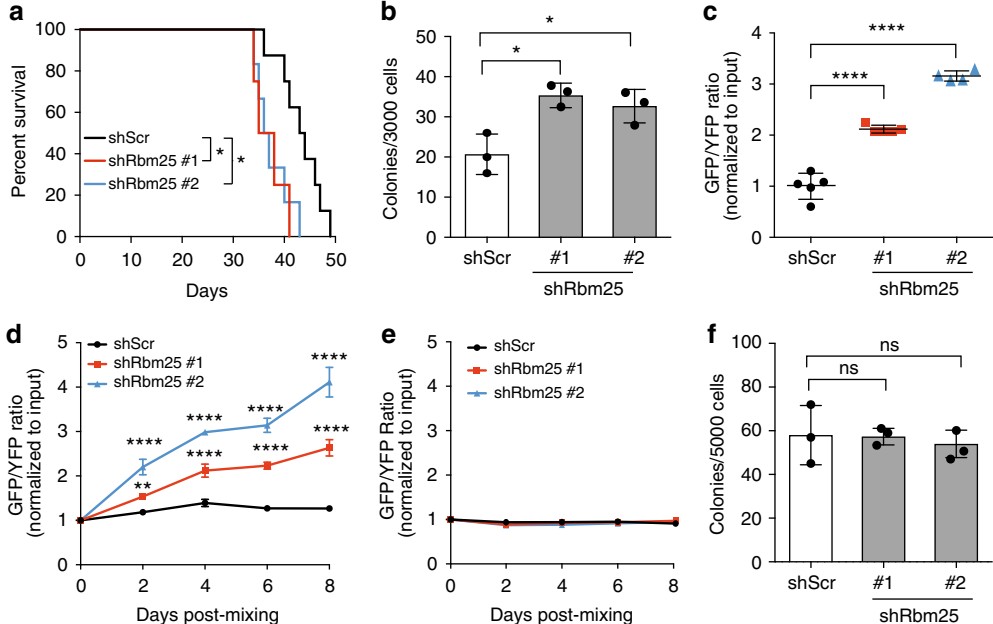

**Fig. 2** Functional validation of *Rbm25* in murine AML. **a** Kaplan−Meier survival analysis of mice injected with AML cells transduced with the indicated shRNAs (*n* = 8, log-rank test, *P < 0.05). **b** Colony-forming assay performed on Lp30 cells transduced with shScr or shRbm25. 3000 GFP⁺ cells were sorted and plated in MethoCult M3434 medium 2 days after transduction (data represent the mean ± s.d., *n* = 3). **c** The effects of shRbm25 on tumor growth in a serial in vivo competitive BM transplantation assay (data represent the mean ± s.d., *n* = 4). **d**, **e** Competitive assays performed on **d** MLL-AF9 and **e** c-Kit⁺ BM cells transduced with shScr or two individual shRNA targeting *Rbm25*. Graph shows the GFP/YFP ratio over time normalized to input mixture. **f** Colony-forming potential of normal c-Kit⁺ BM cells transduced with shScr or shRbm25. Cells were plated in M3434 semi-solid medium 2 days after transduction (data represent the mean ± s.d., *n* = 3). Data were subjected to an unpaired *t* test, and asterisks indicate the following: *P < 0.05; **P < 0.01; ****P < 0.0001. Panels (**b**), (**d**), (**e**) and (**f**) show representative results of two independent experiments. All replicates in (**b**)−(**f**) are biological replicates

in vivo screen. KD of *Sfrs12ip1* and *Prpf40a* displayed similar trends, albeit not with significant FC differences. Of the five depleted hits, *Cpsf6* and *Hnrnpab* showed strong depletion as observed in the screen (Fig. 1f). However, in contrast to what we observed in the c-Kit⁺ screen, in vitro competitive assays on normal c-Kit⁺ cells (Supplementary Fig. 1f) demonstrated that KD of the two targets also led to a gradual depletion of normal hematopoietic cells over time (Supplementary Fig. 1g). Therefore, these candidates were not selected for further studies.

In summary, we have used an in vivo shRNA screening approach to identify two splicing factors with tumor suppressive characteristics in the murine Lp30 AML model.

**Rbm25 KD accelerates leukemogenesis in murine AML.** Among our validated tumor suppressor hits, *Rbm25* KD yielded the most prominent effect on Lp30 AML cells compared to c-Kit⁺ control cells (Fig. 1d, e, see Supplementary Fig. 1h and i for the KD efficiency of all four shRNAs in the library), and we therefore forwarded RBM25 for functional analysis. *RBM25* is ubiquitously expressed in the entire myeloid lineage (Supplementary Fig. 1j and k) and encodes an RNA-binding protein involved in pre-mRNA processing[19]. The protein has previously been implicated in regulation of apoptosis by affecting the balance between the anti- and pro-apoptotic transcripts of the *BCL2L1* gene encoding the key apoptotic regulator BCL-X[20,21]. Yet any role for RBM25 in a leukemic context has not been studied, thus making it a valid focus of further investigation.

We first assessed if *Rbm25* KD impacted on survival of mice transplanted with Lp30 AML cells. To this end, we transplanted 10,000 FACS sorted GFP⁺ Lp30 cells into irradiated recipients and indeed found that mice transplanted with leukemic cells expressing *Rbm25* shRNAs lived significantly shorter than control

mice (36.5 days vs. 43.5 days; Fig. 2a). Furthermore, *Rbm25* KD cells led to increased colony formation in semi-solid media (Fig. 2b).

To test if the effects of *Rbm25* KD were maintained during serial transplantation, we carried out a competitive transplantation experiment with secondary donor cells (Supplementary Fig. 2a). As for the primary transplantations, we observed a strong enrichment of *Rbm25* KD cells vs. competitors (Fig. 2c), thus confirming an in vivo role of *Rbm25* KD in Lp30 AML.

To rule out that the tumor suppressive effect of RBM25 was restricted to the Lp30 AML model, we carried out an in vitro competitive (GFP/YFP) assay using a murine *MLL-AF9* fusion-driven AML model[22]. Coculture of the mixed cells resulted in a 2–4-fold enrichment of *Rbm25* KD cells compared to control cells over an 8-day period (Fig. 2d). Importantly, we did not observe any growth advantage in normal c-Kit⁺ progenitor cells following *Rbm25* KD (Fig. 2e) and no difference was observed in their colony-forming unit (CFU) capacity (Fig. 2f). Efficient *Rbm25* KD in all cellular contexts was confirmed both at mRNA and protein levels (Supplementary Figure 2b and c).

In conclusion, our results demonstrated that reduced expression of RBM25 can potentiate growth of at least two murine AML subtypes. This effect is absent in normal hematopoietic progenitor cells and thus demonstrates that RBM25 exerts a tumor suppressive function in murine AML.

**RBM25 restricts growth of human leukemic cell lines.** Having shown that RBM25 acts as a tumor suppressor in murine AML, we now wanted to test if we could expand this to a human setting. To do so, we first transduced the U937 cell line (cells with myeloid characteristics and unknown mutational status derived from a patient with histiocytic lymphoma) with lentiviral shRNA

vectors targeting *RBM25* and tested the effects on proliferation (Fig. 3a, b). Knockdown of *RBM25* accelerated growth both when assayed in terms of the accumulation of cell numbers (Fig. 3b) and in a label-retaining assay following incubation with carboxyfluorescein diacetate succinimidyl ester (CFSE) (Fig. 3c and Supplementary Fig. 3a). Moreover, when we plated U937 cells in methylcellulose, *RBM25* KD increased their CFU potential and promoted the formation of larger colonies (Fig. 3d). The increase in colony numbers could be rescued by ectopic expression of an shRNA-resistant variant of *RBM25*, demonstrating that the pro-proliferative effects of *RBM25* KD were not caused by off-targets effects (Fig. 3e, f). To further validate the growth inhibitory effects of RBM25, we used CRISPRi-mediated gene silencing in which the noncutting dCAS9-KRAB fusion protein is targeted to the *RBM25* promoter using specific guide RNAs. Consistent with the shRNA KD phenotype, CRISPRi-mediated gene silencing of *RBM25* increased the proliferation of U937 cells in suspension as well as their CFU potential (Supplementary Fig. 3b–d).

In order to gain further insights into the cellular events underlying the growth advantage following downregulation of RBM25, we analyzed the cell cycle status and apoptotic profiles of control and *RBM25* KD U937 cells. Specifically, we found that *RBM25* KD decreased the number of cells in the G0/G1 phase, whereas more cells accumulated in the S/G2/M phase (Fig. 3g). This indicated that cell cycle transition was accelerated upon *RBM25* KD, a phenomenon accompanied by a dramatic reduction in the fraction of Annexin V + apoptotic cells (Fig. 3h). In addition, *RBM25* KD in the t(8;21) AML cell line, Kasumi-1, resulted in similar proliferation and apoptotic phenotypes as those observed in U937 cells (Supplementary Fig. 3e–g)

Overall, these data demonstrate that the tumor-suppressive function of RBM25 that we originally observed in murine AML models can be extended to at least two distinct human models and identifies RBM25 as cross-species regulator of proliferation and apoptosis.

**RBM25 maintains distinct gene expression programs**. In order to gain mechanistic insights into RBM25 function, we next performed RNA-sequencing (RNA-seq) on U937 cells transduced with scrambled or *RBM25*-specific shRNAs and obtained a prominent separation between the transcriptional profiles of the two groups (Supplementary Fig. 4a). Overall, we found 2718 differentially expressed genes with equal numbers of up- and down-regulated genes (adjusted $q < 0.05$; Supplementary Data 2). To explore the potential perturbation of specific molecular pathways after *RBM25* KD, we applied gene set enrichment analysis (GSEA)[23]. Gene sets associated with loss of stemness (in either leukemic or normal hematopoietic stem cells), myeloid differentiation and apoptosis were depleted in *RBM25* KD cells, whereas the opposite was true for MYC targets genes and gene sets implicated in cell cycle progression (Fig. 4a). The deregulation of genes associated with cell cycle and apoptosis support the phenotypic analysis in both human and murine cells (Fig. 3g, h).

As described above, the GSEA suggested that RBM25 may impact on the balance between stemness/self-renewal on one hand and differentiation on the other. In order to directly address this possibility, we took advantage of the recently described 8227 cell culture system, derived from a relapse AML patient. These cells maintain a leukemic hierarchy in culture in which LSCs, progenitors, and blasts can be identified based on CD34 and CD38 expression[24]. 8227 cells were transduced with either *RBM25*-targeting shRNA or scrambled control, and the relative distribution of the CD34⁺CD38⁻ (LSC), CD34⁺CD38⁺ (progenitor) and CD34⁻ (blast) populations was assayed. In line with

the GSEA, downregulation of RBM25 led to significant expansion of the LSC and progenitor compartments at the expense of mature cells (Fig. 4b). In concordance with our observations in U937 cells, the proportion of cells in the S/G2/M phase increased within the two immature populations, while the opposite was the case for cells in the G0/G1 phase (Fig. 4c and Supplementary Fig. 4b). The pro-differentiation effect of RBM25 was also supported by analysis of murine Lp30 AML samples in vivo, where we observed a significant increase in the c-Kit (hematopoietic stem- and progenitor marker) to Mac1 (myeloid differentiation marker) ratio in engrafted BM Lp30 cells expressing low levels of *Rbm25* (Supplementary Fig. 4c).

In summary, the gene expression analysis confirms the observed cell cycle accelerating and anti-apoptotic phenotypes of *RBM25* KD cells. Furthermore, RBM25 appears to restrict the self-renewal and/or promote the differentiation of immature leukemic cells which is consistent with the tumor suppressive function of the protein.

**RBM25 regulates the splicing of specific pre-mRNAs**. Although the role of RBM25 as a splicing regulator is relatively uncharacterized, it has previously been demonstrated to regulate the balance between the pro- and anti-apoptotic functions of BCL-X by affecting splicing of the *BCL2L1* pre-mRNA[20]. In line with this known function of RBM25, we indeed find that the reduction in apoptosis of *RBM25* KD U937 cells is accompanied by changes in the levels of the pro- and anti-apoptotic *BCL2L1* isoforms, effectively changing the balance in favor of apoptotic prevention (Fig. 5a). RNA immunoprecipitation (RIP) experiments demonstrated a selective interaction between RBM25 and the *BCL2L1* transcript, which desensitized cells to ABT-263 (Navitoclax), a selective inhibitor of BCL2 family proteins (Fig. 5b, c). Finally, when we scrutinize an RNA-seq AML dataset from The Cancer Genome Atlas (TCGA)[25], we find that low levels of RBM25 favor a relative high expression of the anti-apoptotic *BCL2L1* isoforms (Fig. 5d). This suggests that RBM25 plays a role in regulating BCL-X function in primary AML patients and that *RBM25* expression levels may be used to predict response to Navitoclax and other BH3-mimetics in AML.

While the findings described above mechanistically explain parts of the *RBM25* KD phenotype, we suspected that RBM25 may also control the levels and/or functions of other regulators of cell proliferation and cell death. In an attempt to uncover these, we subjected our RNA-seq data to the previously reported analytical pipeline SpliceR[26] with the aim of identifying alternatively spliced isoforms. This led to the identification of 338 significantly deregulated transcripts and among these, we noted a prominent increase in the expression of the exon 12 including variant of *BIN1* following KD of *RBM25* (Fig. 6a and Supplementary Data 3). Quantification of the SpliceR output confirmed that *RBM25* KD promoted a marked increase in the relative contribution of the *BIN1* (+12) isoform to the total *BIN1* mRNA pool which also increased (Fig. 6b and Supplementary Fig. 5a). The RNA-seq data were further corroborated by RT-PCR (Fig. 6c), and expression of the BIN1(+12) isoform at the protein level was confirmed by western blotting analysis (Fig. 6d).

BIN1 acts as tumor suppressor through its ability to inhibit MYC activity[27]; however, the inclusion of exon 12 leads to the introduction of an extra domain, which negatively affects the BIN1−MYC interaction via intra-protein steric hindrance[28,29]. To assess whether the effects of *RBM25* KD on cell proliferation are regulated directly by altered BIN1(+12) expression, we specifically knocked down *BIN1*(+12) expression using an shRNA targeting the exon 12 sequence, without a marked effect on the isoforms lacking this exon (Supplementary Fig. 5b).

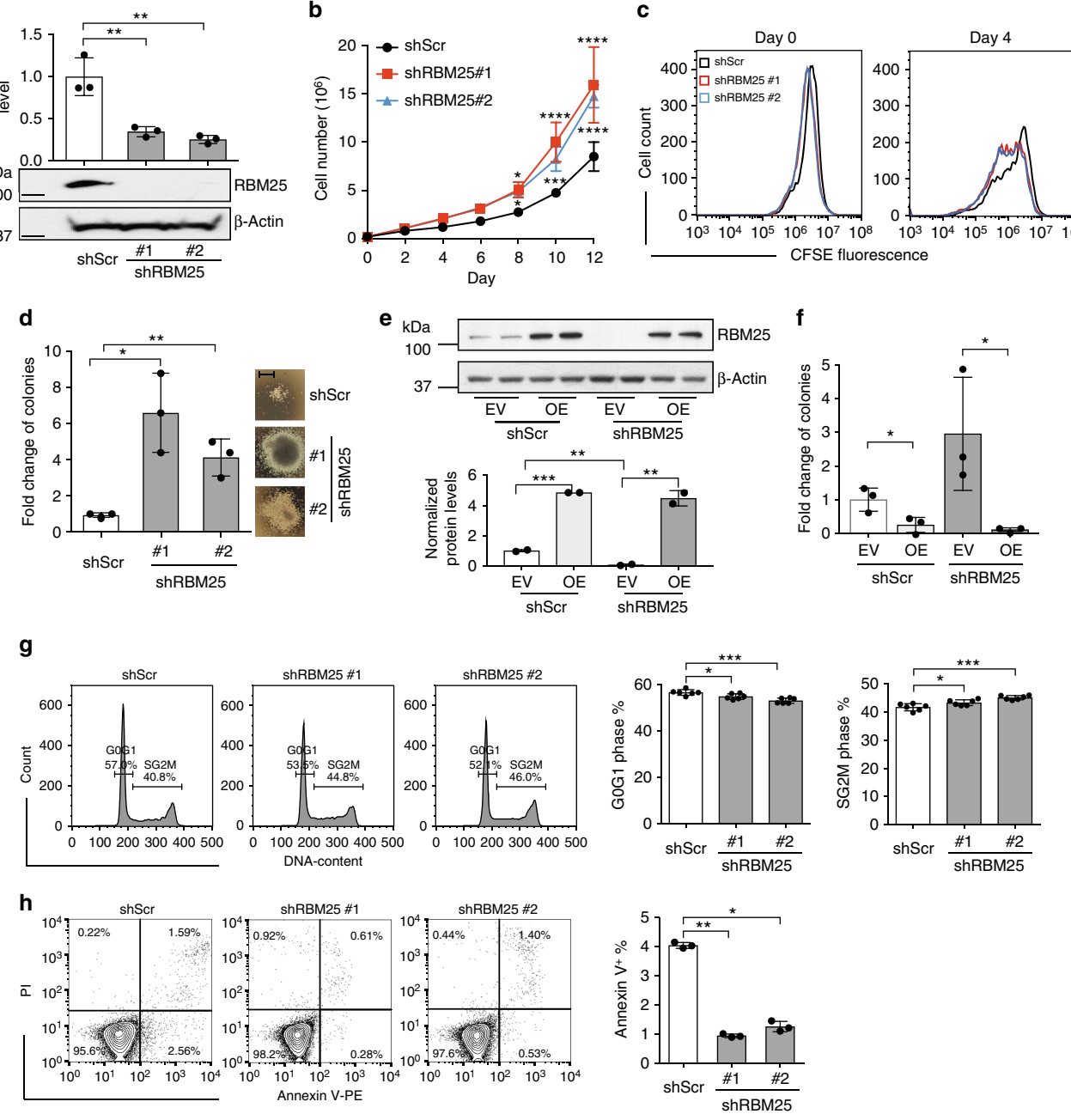

**Fig. 3** RBM25 restricts the growth of human leukemic cells. **a** *RBM25* KD efficiency in U937 cells with two *RBM25* specific shRNAs (shRBM25#1 and shRBM25#2), as analyzed by qPCR (upper panel, data represent the mean ± s.d., *n* = 3) and western blot (lower panel). **b** Proliferation of U937 cells transduced as in (**a**) (data represent the mean ± s.d., *n* = 3). **c** CFSE retention in U937 cells transduced as in (**a**). **d** Colony-forming potential of U937 cells transduced as in (**a**) (data represent the mean ± s.d., *n* = 3). Representative colony morphology is shown next to the graph. Scale bar represents 200 μm. **e** Overexpression of RBM25 in U937 cells with shRNA resistant analyzed by western blot (upper panel). Relative band intensities were quantified by ImageStudioLite and normalized to EV + Scr (lower panel). **f** Rescue of the effect of *RBM25* KD on colony-forming capacity by overexpression of RBM25 (data represent the mean ± s.d., *n* = 3). **g** Cell cycle analysis of U937 cells transduced as in (**a**) assessed by PI staining. Left panel shows representative FACS histograms of cell cycle analysis whereas right panel shows quantitative summary of the G0G1 and SG2M phases (data represent the mean ± s.d., *n* = 6). **h** Apoptosis assessed by flow cytometry using Annexin V/PI staining. Left panel, representative FACS histograms of the cell cycle analysis. Bar plots on the right are quantifications of the Annexin V[+] cells after RBM25 KD (data represent the mean ± s.d., *n* = 3). Data were subjected to an unpaired *t* test and asterisks indicate the following: *$P < 0.05$; **$P < 0.01$; ***$P < 0.001$; ****$P < 0.0001$. All panels show representative results of ≥2 independent experiments. All replicates in (**a**), (**b**), (**d**), (**e**), (**f**), (**g**), and (**h**) are biological replicates

Strikingly, CFU assays revealed that KD of the *BIN1*(+12) isoform almost completely rescued the previously demonstrated effect of *RBM25* KD (Fig. 6e). Moreover, the *BIN1* mRNA transcripts could be specifically immunoprecipitated by an RBM25 antibody suggesting that RBM25 represses the inclusion of exon 12 sequences via direct interactions with *BIN1* pre-mRNA (Fig. 6f).

Taken together, we showed that RBM25 controls the functional properties of not only known targets (BCL-X) but also previously uncharacterized targets such as BIN1 via an alternative splicing

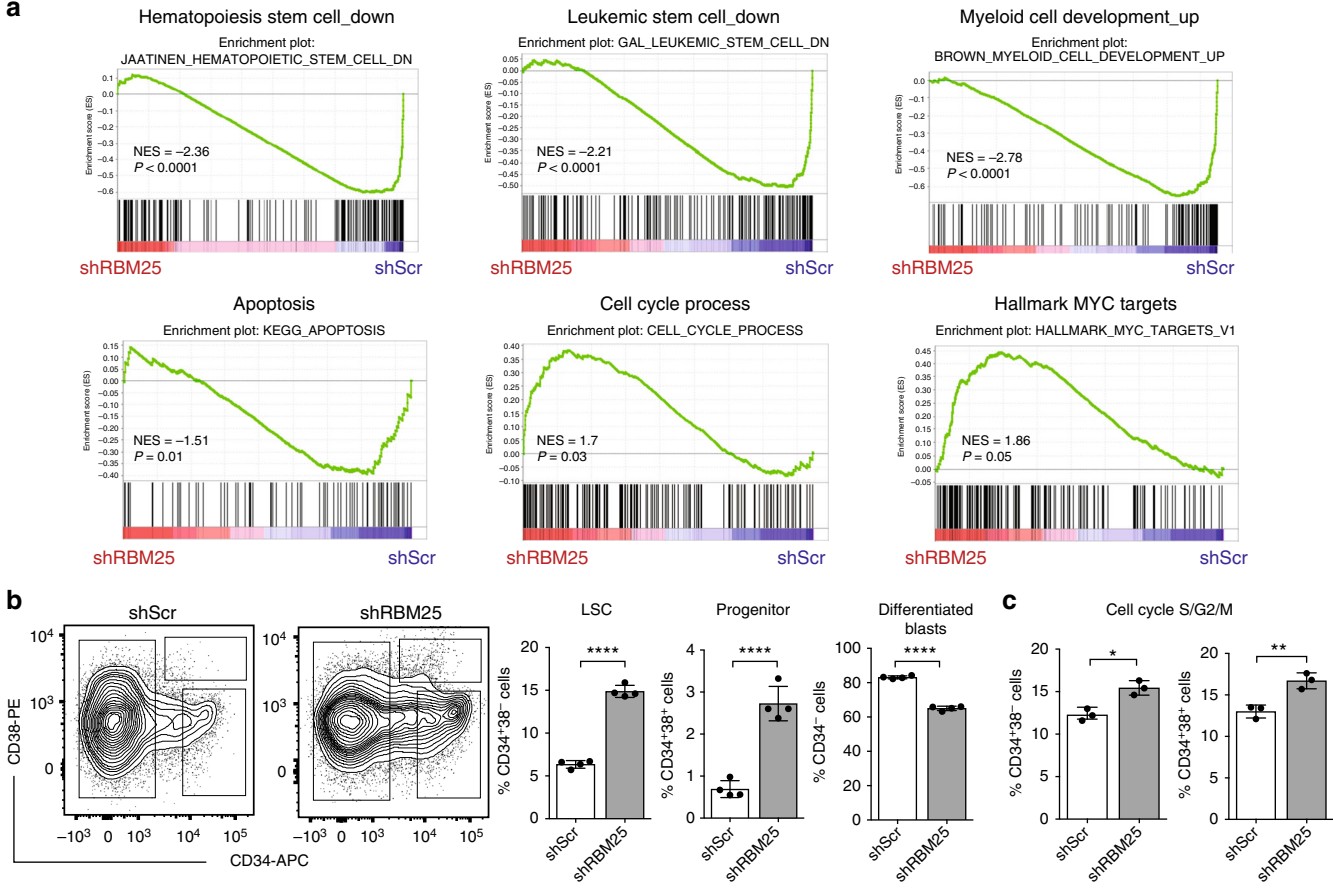

**Fig. 4** *RBM25* knockdown alters the proliferation and differentiation status of human AML cells. **a** GSEA plots showing enrichment of the indicated gene sets in *RBM25* KD vs. control U937 cells. **b** Representative flow cytometry plots depicting changes in CD34$^+$ and CD38$^+$ levels upon *RBM25* KD and quantification of the relative representation of LSCs (CD34$^+$CD38$^-$), progenitors (CD34$^+$CD38$^+$) and differentiated blasts (CD34$^-$) after RBM25 *KD* in human 8227 AML cells (data represent the mean ± s.d., $n = 4$). **c** Cell cycle analysis on indicated populations of 8227 cells following *RBM25* KD as assessed by PI staining (data represent the mean ± s.d., $n = 3$). Data were subjected to an unpaired *t* test and asterisks indicate the following: *$P < 0.05$; **$P < 0.01$; ****$P < 0.0001$. Panels (**b**) and (**c**) show representative results of two independent experiments. All replicates in (**b**) and (**c**) are biological replicates

mechanism. Importantly, targeting the *BIN1* exon 12 variant rescued the *RBM25* KD phenotype.

**RBM25 knockdown activates MYC targets through BIN1**. MYC is a key oncogene that controls several cellular pathways relevant to cancer, including cell cycle entry, ribosome synthesis, and various metabolic processes. Indeed, overexpression of MYC is one of the most common drivers of many cancer types[30], including AML[31]. As shown above, KD of *RBM25* de-repressed an alternatively splicing event and resulted in the inclusion of exon 12 in the *BIN1* pre-mRNA and KD of this variant rescued the *RBM25* KD phenotype. Since BIN1(+12) has been shown to lack the ability to interact with MYC[27,29,32], we hypothesized that the MYC pathway may be activated and that this potentially could underlie the *RBM25* KD phenotype. In accordance with the known posttranslational effect of BIN1 on MYC function[33], we found that MYC protein levels were unchanged following *RBM25* KD (Fig. 7a). As reported above MYC targets were upregulated upon *RBM25* KD (Fig. 4a). In particular, we detect a notable increase in the expression of genes associated with cell cycle (*CDK4*, *CDK6*), protein synthesis (*EIF2A*, *EIF4E2*), energy metabolism (*LDHA*), and DNA metabolism (*CAD*), all of which functionally cooperate to promote cell growth[34]. The de-regulation of these genes was validated both by shRNA and CRISPRi-mediated downregulation of RBM25 (Fig. 7b and Supplementary Fig. 5c) and their MYC-dependency verified by *MYC*

KD (Fig. 7c). Importantly, we showed that KD of the *BIN1*(+12) isoform also partially reduced the expression of the selected *MYC* target genes, thus demonstrating that the effect of *RBM25* KD is mediated by the BIN1(+12) isoform (Fig. 7d).

Next, we wanted to assess how the RBM25-BIN1 pathway affected MYC function. The observed *RBM25* KD-mediated increase in the expression of MYC target genes, in the absence of changes in MYC levels, is consistent with a model where RBM25 levels modulated the transcriptional activity of MYC through the control of BIN1 isoform distribution. To test this model, we cotransfected an MYC-responsive transcriptional reporter construct into U937 cells and demonstrated robust MYC-driven reporter expression which could be abrogated by mutation of the MYC binding sites. The MYC-responsive reporter expression was indeed increased by *RBM25* KD, and this increase was partly suppressed by the *BIN1*(+12)-specific shRNA (Fig. 7e). Finally, we showed that the increased MYC transcriptional activity induced by *RBM25* KD facilitated a modest but significant increase in the tolerance of U937 cells to 10058-F4, an inhibitor of the MYC−MAX interaction (Fig. 7f).

Collectively, we have shown that *RBM25* KD promotes inclusion of exon 12 of the *BIN1* pre-mRNA. This BIN1(+12) isoform acts as a dominant-negative variant leading to increased MYC transcriptional activity which in turn accelerates leukemic proliferation. These findings support a model where RBM25 controls MYC transcriptional activity via BIN1.

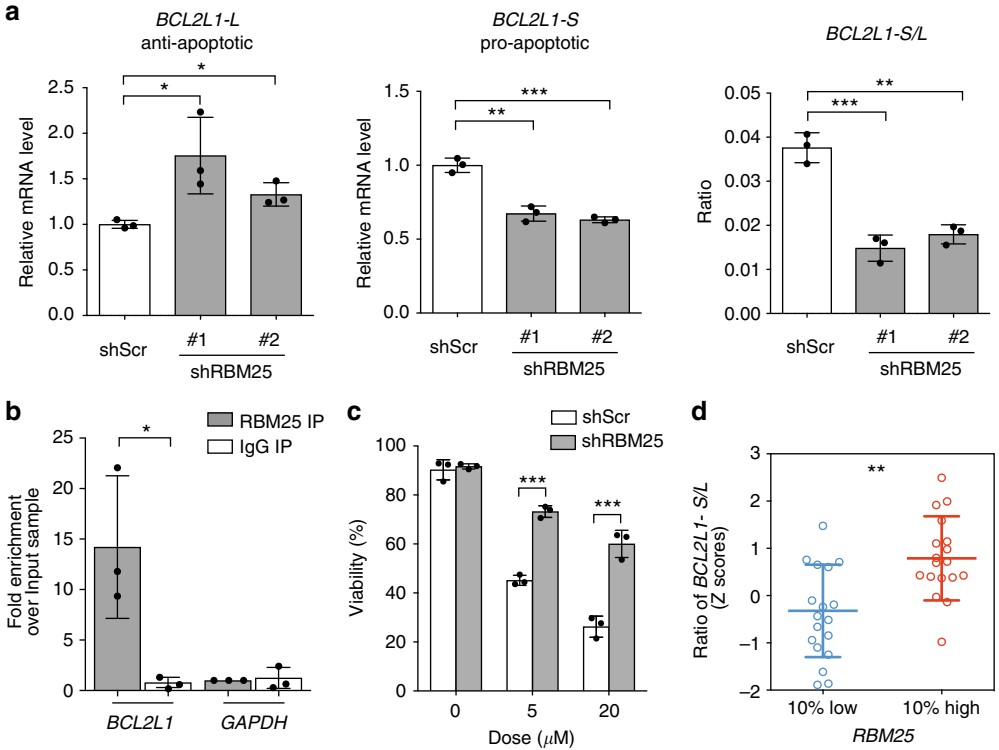

**Fig. 5** *RBM25* knockdown alters the abundance of different *BCL2L1* isoforms. **a** qPCR showing altered expression of *BCL2L1-L* (left), *BCL2L1-S* (middle), and *BCL2L1-L* to *BCL2L1-S* ratios (right) in U937 cells before and after *RBM25* KD **b** RIP assay demonstrating direct interaction between RBM25 and endogenous *BCL2L1* transcripts in U937 cells using GAPDH as a negative control. **c** The effect of ABT-263 on cell viability before and after *RBM25* KD. **d** Correlation between *RBM25* levels and *BCL2L1-L* to *BCL2L1-S* ratios in AML patients (TCGA data[21]). Bar plots in (**a**)−(**c**) represent the mean ± s.d., n = 3. Data were subjected to an unpaired *t* test, and asterisks indicate the following: *$P < 0.05$; **$P < 0.01$; ***$P < 0.001$. **d** The data were Z-score normalized and *P* values were calculated using the Mann−Whitney test (n = 18), **$P < 0.01$. Panels (**a**)(**c**) show representative results of ≥2 independent experiments. All replicates in (**a**)−(**c**) are biological replicates

**RBM25 as a prognostic factor in AML**. Our results so far have uncovered a tumor-suppressive function of RBM25 in murine AML and in human cell lines. To determine if RBM25 also influenced AML patient outcome, we evaluated the prognostic value of *RBM25* mRNA expression using a TCGA RNA-seq AML dataset and associated patient survival data[25]. Notably, we found that low *RBM25* levels were associated with significantly worse overall survival (OS), consistent with a tumor-suppressive function in human AML (Fig. 8a). To investigate whether low *RBM25* levels are associated with particular AML subtypes, we carried out the aforementioned analysis on specific mutations belonging to previously defined subclasses[3] in patients displaying the 10% lowest vs. highest levels of *RBM25* (Supplementary Figure 6a). This analysis revealed that low-expressing patients displayed a lower mutational load and a complete absence of the APL-associated t(15;17) translocation. Both observations fit well with our demonstration that *RBM25* KD results in accelerated tumor progression and delayed myeloid differentiation. It is tempting to propose that the reduced proliferation associated with high *RBM25* levels necessitates additional mutational events to sustain AML. Conversely, the t(15;17) translocation is associated with favorable prognosis[3] and a relatively differentiated phenotype, thus explaining the observed mutual exclusion of low *RBM25* expression and the t(15;17) translocation.

To address a possible link between *RBM25* and *BIN1*(+12) expression in AML patients, we next assessed the expression of *BIN1*(+12) in patients displaying the 10% highest vs. lowest *RBM25* levels. Consistent with our functional data, this analysis demonstrated an inverse correlation between *BIN1*(+12) and

*RBM25* (Fig. 8b). Finally, we observed an inverse correlation between the expression of RBM25 and an MYC target signature (Fig. 8c), consistent with our in vitro data demonstrating that *RBM25* KD facilitates the upregulation of an MYC-driven transcriptional program.

Finally, to address whether a concomitant decrease in *RBM25* expression and activation of MYC targets is associated with specific AML subtypes we partitioned the TCGA dataset into two distinct groups: Group 1 contains patients which display low expression of RBM25 (low 25% percentile) and high level of MYC target gene expression (MYC target gene score > median) and Group 2 containing patients with the opposite characteristics. We subsequently assessed which genetic lesions were differentially represented between them (Supplementary Fig. 6b). This analysis revealed that mutations in epigenetic modifiers and *NPM1* were overrepresented (albeit not significantly for the latter) in samples with low *RBM25* expression and high MYC score, whereas the opposite was the case for t(15;17) and t(8;21) translocations. The trend towards overrepresentation of *NPM1* mutations in samples with low *RBM25* expression and high MYC score is potentially interesting, since the *NPM1* gene has previously been shown to be a direct MYC target[35,36]. It is tempting to speculate that an increased *trans*-activation potential of MYC resulting from low *RBM25* expression may promote increased NPM1 levels which may, in turn, provide a selective pressure for the acquisition of *NPM1* LOF mutations.

Collectively, these data suggest that the RBM25−BIN1−MYC axis is of functional importance not only in murine AML and human leukemic cell lines, but also in primary human AML.

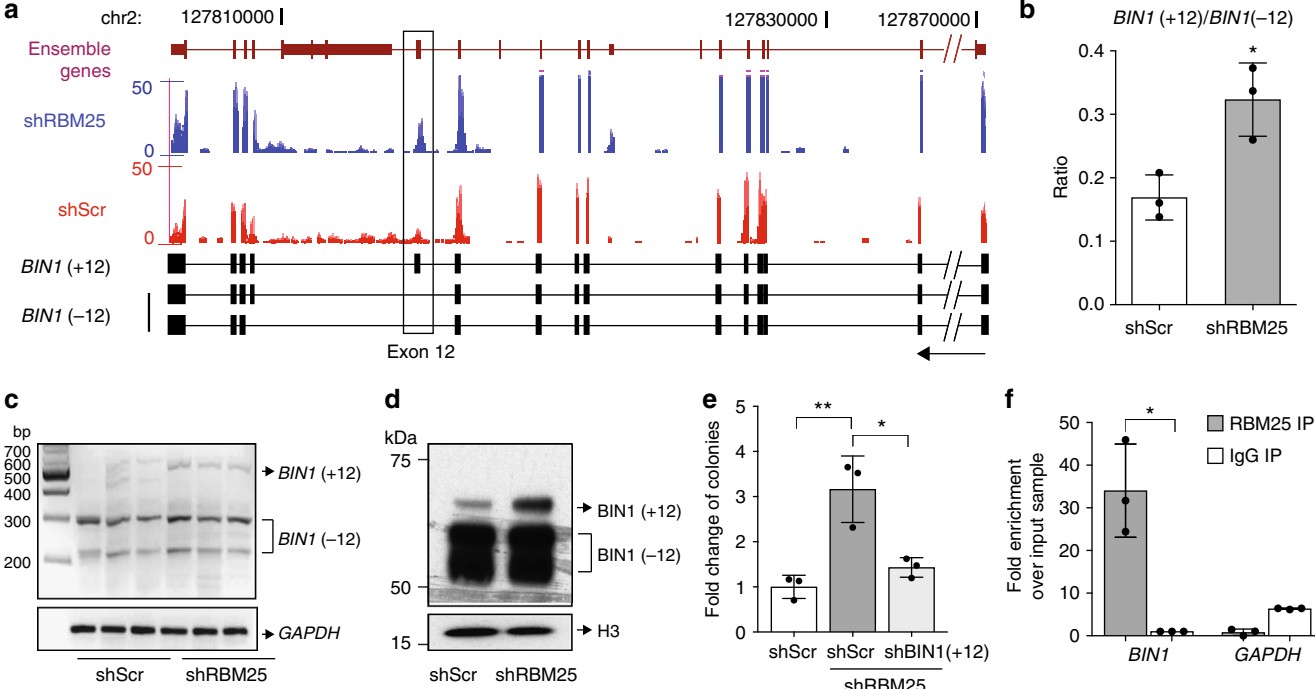

**Fig. 6** *RBM25* knockdown affects alternative splicing of the *BIN1* pre-mRNA. **a** Genome browser tracks showing RNA-seq reads coverage across the *BIN1* locus in shScr control (red) and *RBM25* KD (blue) U937 cells (upper panel). Lower panel illustrates three alternatively spliced isoforms of the *BIN1* pre-mRNA. The black rectangle highlights the *BIN1* exon 12. **b**−**d** *RBM25* KD alters the relative usage of BIN1 isoforms as assessed by **b** RNA-seq, **c** semi-quantitative RT-PCR, and **d** western blot analyses. **e** Partial rescue of the increased colony-forming capacity of *RBM25* KD cells by specific KD of BIN1(+12). **f** RNA immunoprecipitation (RIP) demonstrating specific binding of RBM25 to *BIN1* mRNA. Bar graphs in (**b**), (**e**) and (**f**) represent the mean ± s.d., $n = 3$, and *P* values were determined by *t* test. *$P < 0.05$, **$P < 0.01$. Panels (**b**), (**d**) and (**f**) show representative results of ≥2 independent experiments. All replicates in (**b**), (**c**), (**e**) and (**f**) are biological replicates

## Discussion

Pre-mRNA splicing is a key process with the potential to impact on all cellular pathways. Recent cancer sequencing efforts have uncovered splicing factors as being frequently mutated in hematological malignancies, including MDS and AML, and pre-mRNA splicing is frequently deregulated in these diseases. Here, we have used an in vivo shRNA screening approach in a murine AML model in order to identify novel splicing regulators which act as tumor-promoting or -suppressing factors in AML. Among several validated candidates, we focused on RBM25 which we demonstrated to have tumor-suppressive properties in two murine AML models, two human myeloid cell lines and in the hierarchical organized 8227 AML cell culture system. In contrast, no effects could be demonstrated of knocking down the gene in normal murine hematopoietic cells. Mechanistically, the anti-proliferative effect of RBM25 correlated with delayed cell cycle progression, increased apoptosis and accelerated myeloid differentiation. In addition, RBM25 restrict both CFU and LSC numbers in U937 and 8227 cultures, respectively, suggesting that its main function is to act on the most primitive leukemic cells.

RBM25 is a poorly characterized RNA binding protein that has previously been implicated in specific splicing events in human heart failure and as a crucial factor for plant growth and abiotic stresses in Arabidopsis[37–39]. A recent study using CRISPR/CAS9-mediated knockout demonstrates that complete loss of RBM25 is detrimental to growth of human cancer cell lines and leads to widespread deregulation of pre-mRNA splicing[40]. In the context of our work, this suggests that the impact of RBM25 on proliferation, apoptosis, and splicing is not only highly dose-dependent but may also be cell-type-dependent. Interestingly, recurrently heterozygously mutated splicing factors in hematological malignancies (such as *SRSF2* and *SF3B1*) are associated

with embryonic lethality following their complete deletion in mice[41,42]. Thus, the concept of a (near-) essential splicing regulator having tumor-promoting functions at reduced levels, as reported in the present work, suggest that AML (and perhaps other cancer cells) cells tolerate a reduction in essential splicing function, if these are counteracted by beneficial events such as those reported here. In further support of its tumor suppressor function, RBM25 was previously found to be mutated in breast cancer and to favor the expression of the pro-apoptotic isoform of the *BCL2L1* pre-mRNA, the latter via its role in 5′ splice site selection in the context of the U1 snRNP[20,43,44].

In the present work, we also showed that RBM25 abrogates the inclusion of exon 12 in the *BIN1* pre-mRNA, thus preventing the generation of the dominant-negative BIN1 isoform which is unable to repress MYC activity. Indeed, shRNA-mediated downregulation of the *BIN1(+12)* transcript partly rescued the increase in growth, CFU activity and expression of MYC target genes induced by downregulation of RBM25. Given the well-known pro-proliferative role of MYC in many settings, the increase in MYC activity therefore seems to underlie the accelerated cell cycle progression following *RBM25* KD. Moreover, MYC has also been reported to be important for LSC function which again is consistent with our findings of increased LSC numbers in the 8227 system and the increase in CFU activity in U937 cells following *RBM25* KD[45,46].

Overall, our data suggest a model where the control of MYC activity is a core function of RBM25 and that it is part of molecular rheostat to keep MYC activity under control in normal cells (Fig. 8d). Interestingly, the proposed MYC rheostat is further protected by the impact of RBM25 on splicing of the *BCL2L1* pre-mRNA. Specifically, although MYC is a growth-promoting oncogene, its overexpression in normal cells is associated with

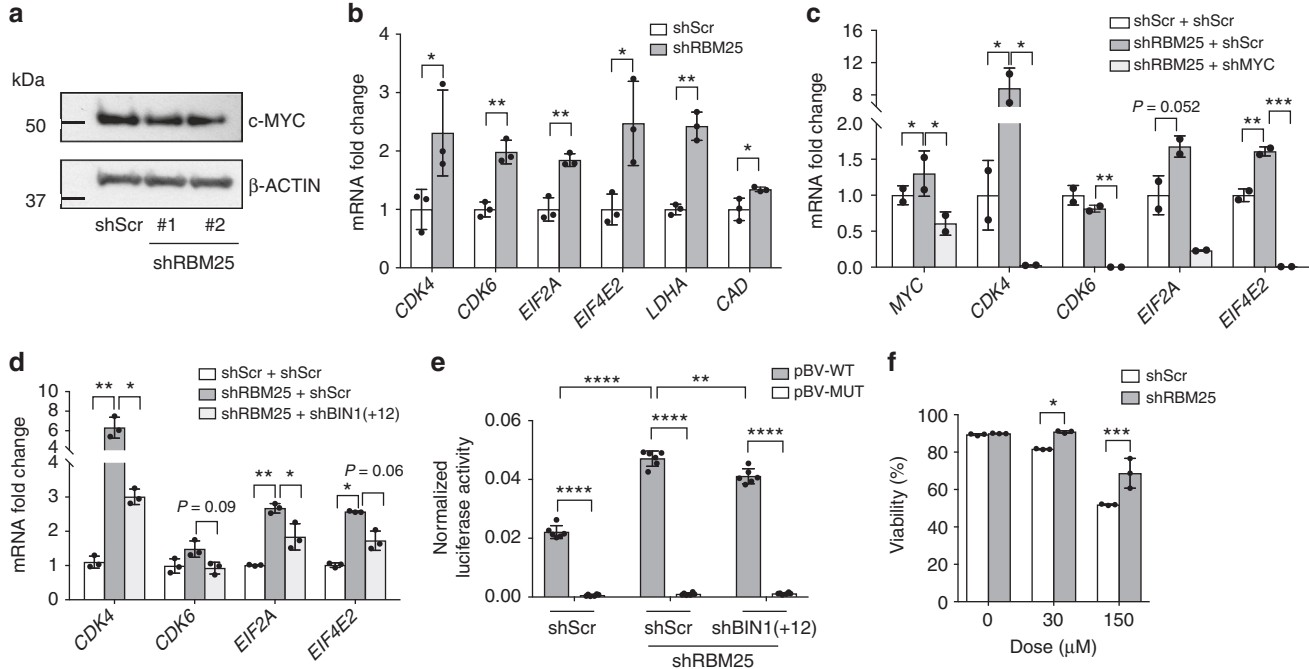

**Fig. 7** RBM25 restricts MYC activity through inhibition of the expression of the BIN1(+12) isoform. **a** Western blot of MYC expression in U937 cells transduced with the indicated shRNAs. **b–d** The expression of MYC target genes as assayed by qPCR in U937 cells after **b** RBM25 KD, **c** double KD of RBM25 and MYC (data represent the mean ± s.d., n = 2, P values were calculated using a one-way ANOVA test), and **d** double KD of RBM25 and the BIN1 (+12) isoform (data represent the mean ± s.d., n = 3). **e** Dual luciferase reporter assay demonstrating the effect of RBM25 KD on a reporter construct containing either WT (pBV-WT) or mutated (pBV-MUT) MYC binding sites with and without simultaneous KD of the BIN1(+12) isoform (data represent the mean ± s.d., n = 6). **f** Effect of the MYC inhibitor 10058-F4 on the viability of U937 cells with or without RBM25 KD (data represent the mean ± s.d., n = 3). Data were subjected to an unpaired t test, and asterisks indicate the following: *P < 0.05, **P < 0.01 ***P < 0.001, ****P < 0.0001. All panels show representative results of ≥2 independent experiments, and all replicates in (**b**)–(**d**) are biological replicates

apoptosis, and MYC-dependent tumors therefore have to disable apoptotic pathways[47]. Thus by favoring the expression of the pro-apoptotic BCL-XS isoform, RBM25 provides the system with a fail-safe mechanism in case of MYC activation by RBM25-independent means. The proposed MYC rheostat may also be important during normal development in order to control MYC activity and thereby avoid overgrowth of particular cell types.

In leukemia, downregulation of RBM25 allows premalignant or malignant cells to harvest the pro-proliferative benefits of MYC (via expression of the BIN(+12) isoform), while at the same time escaping apoptosis (via a shift towards the anti-apoptotic BCL-XL). The proposed MYC rheostat also appears to be of clinical relevance. Specifically, in AML patients, we showed that RBM25 low-expressing patient samples express high levels of the BIN (+12) isoform, high levels of MYC target genes and are associated with a particular poor outcome and specific AML subtypes. Thus, modulation of RBM25 levels or activity may constitute a potential therapeutic option in a subset of AML patients.

Does the RBM25-BIN1-MYC pathway identified in the present work play a role in other cancers? Given that MYC deregulation is a hallmark of many cancers, we hypothesize that at least some of these events are caused by derepression of MYC via the RBM25−BIN1−MYC pathway. Given that this pathway operates at the posttranscriptional levels, these cancers would not display altered levels of MYC transcript but will be characterized by the increased expression of downstream MYC targets. This hypothesis is supported by data from melanoma where tumor development is associated with the expression of BIN(+12) variant which was further demonstrated to be unable to inhibit MYC-mediated malignant transformation[29]. The extent to which this observation can be expanded to other tumor types is currently under investigation.

In summary, we have used an in vivo shRNA screening approach to uncover a novel splicing-dependent mechanism for the control of MYC activity which appears to be operative in primary cancer. To the best of our knowledge, this represents one of the few examples where specific splicing events have been demonstrated to underlie the phenotype resulting from splicing factor deregulation in a relevant disease context. As such our studies suggest that disease-relevant splicing factors expand beyond those that are found to be recurrently mutated in diseases like cancer.

## Methods

**Animal studies**. B6.SJL recipients (all female, 10–15 weeks of age), preconditioned by a sublethal (500 cGy) dose of gamma-irradiation 17 h prior to transplantation, were injected intravenously (IV) with Lp30 AML cells. In order to progress the Lp30 model[18] to an aggressive state with short latency and high penetrance, we further enriched for LSCs activity by serial transplanting leukemias through secondary and tertiary recipients.

Competitive BMT assay: $10^5$ transduced target-specific-shRNA-GFP-positive cells were mixed in a 1:1 ratio with shScr-YFP-positive cells and transplanted into sublethally irradiated recipients who were sacrificed after 4 weeks. For serial competitive BMT assay, shRbm25-GFP and shScr-YFP (in the pMLS backbone)-positive cells were harvested from the primary transplanted CD45.2+ mice 3 weeks after injection and mixed in a 1:1 ratio. Fifty thousand cells were transplanted into sublethally irradiated recipient CD45.1+ mice who were sacrificed and assayed after 3 weeks.

Survival assay: Sublethally irradiated recipients were transplanted with 10,000 transduced and sorted GFP+ Lp30 cells along with $4.5 \times 10^5$ cells/mouse irradiated (2000 cGy) BM cells as support.

In all experiments, recipient mice were randomly assigned to receive different experimental transplants and mice were shuffled between cages. No blinding was performed. Control and test groups contain 4−8 mice each and group size was not predetermined. No animals were excluded from analysis. Mice were monitored on a daily basis for disease progression and sacrificed when they became moribund. All mouse experiments were conducted according to protocols approved by the Danish Animal Ethical Committee.

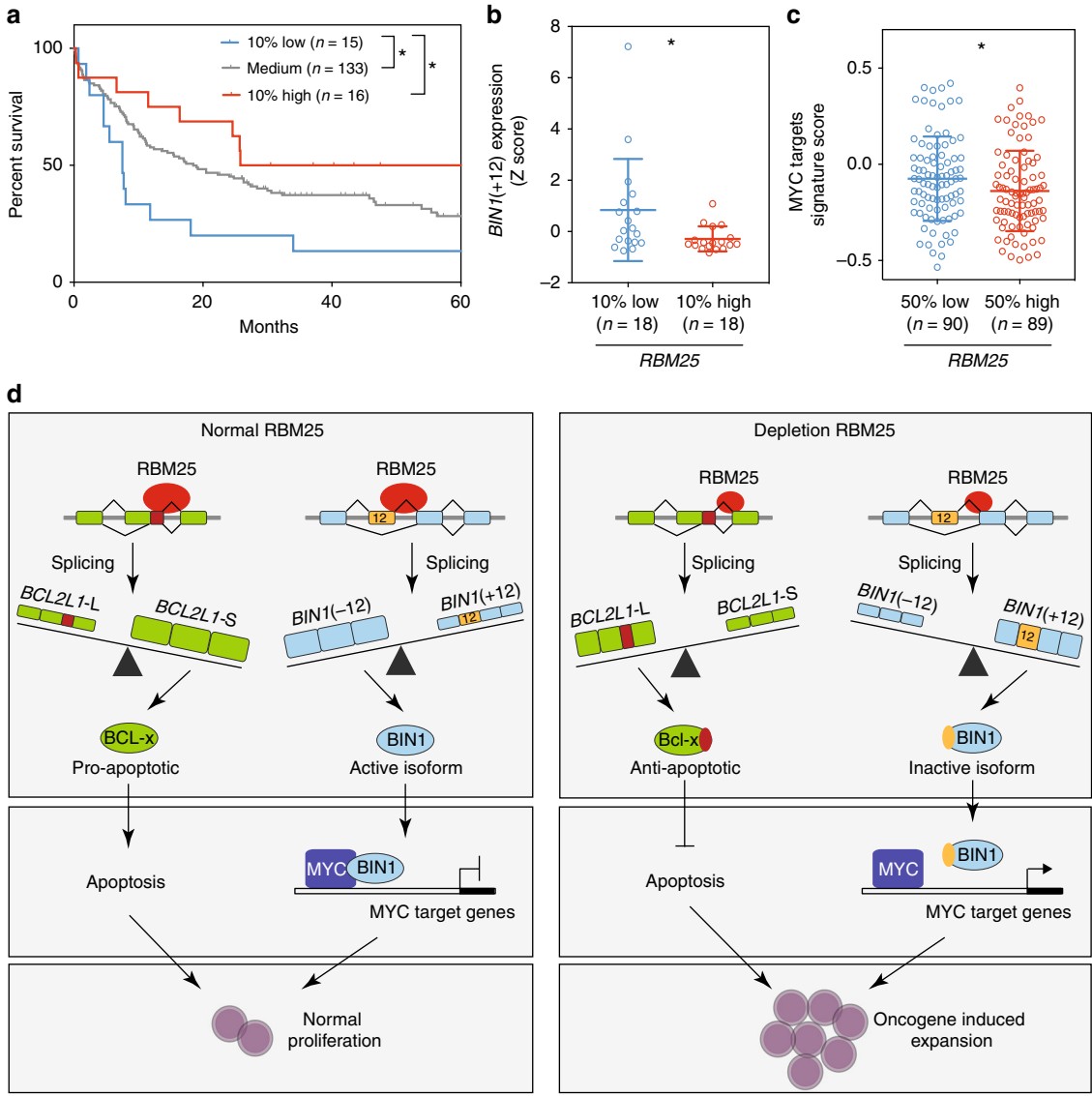

**Fig. 8** High *RBM25* levels correlate with prolonged patient survival, alternative *BIN1* mRNA splicing, and MYC target gene repression. **a** Overall survival of AML patients (TCGA dataset) grouped according to *RBM25* expression as indicated. Patients with survival times longer than 60 months were excluded from the analysis. P values were calculated using Log-rank test, *P < 0.05. **b** Correlation between expression of *RBM25* (grouped as in **a**) and the *BIN1*(+12) isoform. The data was Z-score normalized, and P values were calculated using the Mann−Whitney test, *P < 0.05. **c** Correlation between *RBM25* expression and the ALFANO_MYC_target genes signature (MSigDB) score in patient samples. The patients were divided into low (below median) or high (above median) *RBM25* expression groups, and the data were normalized to the corresponding MYC expression for each patient. P values were calculated using the Mann−Whitney test, *P < 0.05. **d** Schematic model depicting the effect of RBM25 on alternative splicing of *BCL2-L1* and *BIN1* mRNA and consequently on MYC function, apoptosis, and cell proliferation

**Generation of splicing factor pMLS library**. In total, 613 shRNAs targeting 230 known or putative splicing factor genes (full list is provided in Supplementary Data 1) in the pGIPZ vector (Dharmacon) were subcloned into pMLS (MSCV-LTRmir30-SV40-GFP)[48] with *Xho*I and *Eco*RI.

**Retrovirus production and transduction**. Retroviral supernatants were generated by CaPO₄-mediated transfection of Phoenix E cells. Forty-eight hours post-transfection, cell supernatants were harvested, filtered through 0.45 μm acrodisc syringe filters (Pall), and frozen for storage. Pooled retrovirus supernatants for screening were generated by pooling equal amounts of the pMLS constructs prior to transfection.

For transduction, retroviral supernatant (800 μl/well) was added onto retronectin (Takara)-coated nontissue culture treated 24-well plates and centrifuged at 2000 × *g* for 50 min at 32 °C. After aspiration of the supernatant, cells were seeded at a density of 1×10⁶/well. The transduction was repeated the following day, and the cells were cultured for 24 h prior to Fluorescence Activated Cell Sorter (FACS) sorting of transduced (GFP⁺) cells.

**Cell culture conditions**. Lp30 AML cells were cultured in X-vivo 15 media (Lonza) supplemented with 50 ng/ml murine stem cell factor (SCF), 50 ng/ml human interleukin-6 (IL-6), 10 ng/ml murine interleukin-3 (IL-3) (all from Prepotech), 1% bovine serum albumin (Stem Cell Technologies), 0.1 mM β-mercaptoethanol (Sigma-Aldrich), 1% L-glutamine (Gibco), and 1% penicillin/streptomycin (pen/strep, PAA).

Wild-type BM cells were enriched for c-Kit⁺ cells by incubation with anti-CD117 microbeads (Miltenyi Biotec) and subsequent separation on MACS LS columns (Miltenyi Biotec). c-Kit⁺-enriched cells were grown in RPMI-1640 medium (Life Technologies) supplemented with 20% fetal bovine serum (Hyclone), 20% WEHI conditioned medium, 20 ng/ml SCF, 10 ng/ml IL-6, and 1% pen/strep. MLL-AF9 cells were derived from a primary MLL-AF9-transduced murine AML[49] and grown using the same conditions as for c-Kit⁺ cells.

For in vitro competitive assays on c-Kit⁺ cells and MLL-AF9 cells, target-specific shRNA-GFP-positive cells were mixed in a 1:1 ratio with corresponding shScr-YFP positive cells and seeded at a density of 600,000 cells/ml. Cells were kept subconfluent and counted and re-seeded every second day.

U937 cells were grown in RPMI-1640, GlutaMAX™ supplement medium (Thermo Fisher Scientific) containing 10% fetal calf serum (FCS) (HyClone) and

1% penicillin/streptomycin (Thermo Fisher Scientific). Kasumi-1 cells were grown in RPMI-1640, GlutaMAX™ supplement medium (Thermo Fisher Scientific) containing 20% FCS (HyClone) and 1% penicillin/streptomycin (Thermo Fisher Scientific).

Early passage 8227 cell cultures were grown in StemSpan SFEM II (Stem Cell Technologies) supplemented with Flt3-L (50 ng/ml), IL-3 (10 ng/ml), SCF (50 ng/ml), IL-6 (10 ng/ml), granulocyte colony-stimulating factor (10 ng/ml), thrombopoietin (25 ng/ml) (all from Miltenyi Biotec SE)[24]. Individual cultures were passaged every 6–7 days and monitored by flow cytometry using CD34 and CD38. U937 and kasumi-1 cells were purchased from the American Type Culture Collection (ATCC). All cell lines used have been tested for Mycoplasma.

**In vivo pooled shRNA screening**. In vivo screening on Lp30 murine AML cells was carried out by transducing Lp30 cells with subpools (150 shRNAs) of shRNA expressing retrovirus. Transduced cells were injected (IV) into sublethally irradiated recipient mice in numbers corresponding to at least 500,000 GFP+ cells (assessed by FACS analysis prior to injection). Recipient mice were sacrificed, and BM cells were harvested 4 weeks after transplantation.

In the in vitro screen carried out on c-Kit+ cells, the entire splicing factor library containing 613 shRNAs was used to transduce two million c-Kit+-enriched BM cells which were kept in culture for 7 days after transduction. The shRNA hairpin region was PCR amplified from purified genomic DNA using primers carrying Illumina adaptors and barcodes (Supplementary Table 3) under the following conditions: 1 μl 10 μM primary PCR primer mix, 1 μl 10 mM dNTP mix, 1 μl 10× HiFi buffer, 0.2 μl of Platinum® Taq DNA polymerase (Invitrogen) and 1 μg genomic DNA in a total reaction volume of 50 μl. Reactions were subjected to 94 °C for 30 s and then run 40 cycles of 94 °C for 30 s, 62 °C for 30 s and 68 °C for 25 s followed by 68 °C for 5 min. PCR products were then gel-purified and subsequently sequenced on Illumina HiSeq 2500 system.

Sequencing reads were de-multiplexed and mapped using bowtie2 [50] onto an shRNA library pseudogenome as previously described[49]. The raw sequencing counts were normalized to the total number of reads for each replicate. The fold change was calculated as the ratio of normalized reads between the two time points, divided by the normalized reads of control shRNAs.

**Colony-forming assays of murine cells**. FACS sorted transduced cells were plated on MethoCult M3434 (Stemcell Technologies). Colonies were scored after 14 days by light microscopy.

**DNA constructs and cloning**. For experiments in human cells, we used a lentiviral vector (pLKO.1-puro, Sigma-Aldrich) expressing shRNA targeting RBM25. An shRNA-resistant RBM25 variant (RBM25-R, with mutation of C1425T, T1428C, G1431A, T1434C, A1440G) was generated from a human RBM25 open reading frame (ORF) pDONR223 vector (Clone ID: BC136775-ORF, TransOMIC) using the QuickChange II XL Site-Directed Mutagenesis Kit (Agilent Technologies) according to the manufacturer's instructions. Successful mutagenesis was confirmed by Sanger sequencing, and the mutant RBM25 ORF was introduced into the lentiviral expression vector pLX304 (Addgene plasmid # 25890) using gateway cloning.

For RBM25 guide RNA (gRNA) cloning (sequences are listed in Supplementary table 4), oligos were phosphorylated, annealed and ligated into BsmBI digested pLKO5.sgRNA.EFS.GFP (Addgene plasmid # 57822).

shRNA targeting BIN1(+12) and MYC (sequences are listed in Supplementary Table 4) were cloned into pLKO.3G (Addgene plasmid #14748). All restriction enzymes were purchased from NEB and used according to the manufacturer's instructions. pHR-SFFV-KRAB-dCas9-P2A-mCherry was from Addgene (plasmid # 60954).

The MYC reporter plasmid pBV-Luc wild-type (wt) MBS1–4 (Addgene, plasmid #16564) contains four tandem MYC binding sites from the human CDK4 promoter located upstream of Firefly luciferase. pBV-Luc mut MBS1–4 (Addgene, plasmid 16565), which has all four MYC binding sites mutated was used as a negative control. pRL-SV 40 (Promega) is a Renilla luciferase-expressing plasmid that was used as an internal control in the dual luciferase assay.

**Lentivirus production and transduction human cells**. Lentivirus was produced by CaPO4-mediated transfection of HEK-293FT cells with 4 μg of pCMV-VSV-G, 8 μg of PAX8 packaging plasmid, and 10 μg of the lentiviral vector plasmid. Production of viral supernatants and transduction was carried out as with retroviral constructs. Cells transduced with pLKO-puro constructs were selected using 2 μg/ml of puromycin (Sigma) for 3 days, whereas pLX304-blasticidin transduced cells were selected using 5 μg/ml blasticidin (Thermo Fisher Scientific) for 5 days.

**Cell growth, CSFE assay, and colony-forming assays of human cells**. Cells were transduced as described above and plated at 100,000 cells/ml for proliferation assays. Cell proliferation was either assessed directly by counting or indirectly by CFSE staining using the CellTrace™ CFSE Cell Proliferation Kit (Thermo Fisher Scientific) according to the manufacturer's protocol. For colony-forming assays, transduced cells were FACS sorted and plated on MethoCult H4434 (Stemcell Technologies). Colonies were scored after 12 days by light microscopy.

**Flow cytometry analysis and cell sorting**. Cell sorting was carried out on FACSAriaI or FACSAriaIII (BD Biosciences), whereas analytical stains were assessed on LSRII or FACSCalibur (BD Biosciences).

Retrovirally transduced murine cells were washed with FACS buffer (3% FCS in phosphate buffer saline (PBS)), and stained with 7-AAD (Invitrogen, 1:1000) to exclude dead cell prior to sorting or analyzing for GFP/YFP expression. For surface marker analysis, BM cells collected from tibia, femur, and ilium were washed with FACS buffer, and stained for 15 min at 4 °C using the following antibody cocktails:

CD45.2-PE-eFlour610 (eBioscience, clone 104, 1:50), CD45.1-PECy7 (eBioscience, clone A20, 1:50), CD11b-APC (eBioscience, clone M70, 1:800), c-Kit-eFlour780 (eBioscience, clone 2B8, 1:200). Gating strategy as in Supplementary Fig. 4c. For human cell lines cell cycle analysis: Cells were fixed, permeabilized, and stained with PI (Invitrogen). For human cell line apoptosis analysis: cells were washed with FACS buffer (3% FCS in PBS), and stained using PE Annexin V apoptosis Detection kit (BD Biosciences), according to the manufacturer's recommendations. For 8227 cells LSC/progenitor/blast population analysis: cells were washed with FACS buffer (3% FCS in PBS), and stained with CD34-APC (BD Biosciences, clone 581, 1:200) and CD38-PE (BD Biosciences, clone HB7, 1:50).

All flow cytometry data were analyzed using the FlowJo 10.1 software (Treestar, Ashland, OR, USA).

**RNA extraction, cDNA synthesis, and qPCR analysis**. RNA was extracted using the AllPrep DNA/RNA Mini kit (Qiagen) following the manufacturer's instruction. cDNA was synthesized by ProtoScript cDNA synthesis kit (New England Biolabs). Real-time PCR reactions were performed on a LightCycler 480 (Roche) using SYBR Green I PCR Master Mix (Roche). Expression was normalized to the housekeeping genes Rpo or Actg1 for murine cells, and GAPDH or H6PD for human cells. All primers used are listed in Supplementary Information as Supplementary Table 5.

**Protein extraction and western blot analysis**. $1 \times 10^6$ cells were collected by centrifugation, washed twice with cold PBS and resuspended in 100 μl of 1× Laemmli Sample buffer (Bio-Rad) with freshly added β-mercaptoethanol (Sigma-Aldrich). The supernatant was collected by centrifugation and boiled for 5 min. Fifteen microliters of the resulting protein lysate was separated by 4–12% SDS-PAGE.

For BIN1 detection, $2.5 \times 10^6$ cells were washed twice with cold PBS, lysed on ice for 10 min with hypotonic buffer (20 mM HEPES (2-[4-(2-hydroxyethyl)piperazin-1-yl]ethanesulfonic acid) pH 8.0, 10 mM KCl, 1 mM MgCl₂, 20% glycerol, and 0.1% Triton-X100) and cleared by centrifugation. The pellets were resuspended in high salt lysis buffer (10 mM Tris-HCl, PH7.9, 420 mM NaCl and 0.1% NP-40), and then sonicated five cycles (15 s on/10 s off) in a Bioruptor (Diagenode). The supernatant nuclear fraction was collected after centrifugation. Protein concentration was measured by Spectrophotometer DS-11 FX (DeNovix) and the protein lysate was separated on 4–12% SDS-PAGE gels.

Protein was transferred to polyvinylidene difluoride membranes, which were blocked in 5% skim milk in tris-buffered saline supplemented with 0.05% Tween 20 and incubated with primary antibodies overnight at 4 °C and hereafter with horseradish peroxidase (HRP)-conjugated secondary antibody (1:4000, Dako), diluted in tris-buffered saline-0.05%Tween20, for 1 h at room temperature. Proteins were detected by enhanced chemiluminescence (Thermo Fisher Scientific). The following antibodies were used: anti-RBM25 (sc-374271, Santa Cruz Biotechnology, 1:1000 dilution), anti-BIN1 (ab182562, Abcam, 1:1000 dilution), anti-c-MYC (sc-40, Santa Cruz Biotechnology, 1:1000 dilution), anti-H3 (ab10799, Abcam) and anti-β-actin (A3854, Sigma-Aldrich). Uncropped scans of all shown western blots are available in Supplementary Fig. 7.

**Transient transfections/dual luciferase assay**. Cells were grown to confluence in six-well plates and transfected with 1 μg of pBV-WT or pBV-MUT (Addgene) and 1 μg of the control pRL-SV40 vector (Promega) using TransIT®-Jurkat Transfection Reagent (MIRUS) according to the manufacturer's recommendations. Twenty-four hours posttransfection, the cells were lysed and assayed using the dual luciferase reporter assay system (Promega) on a Glomax 96 instrument (Promega).

**RNA immunoprecipitation**. $1 \times 10^6$ cells were pelleted by centrifugation at $300 \times g$ for 10 min at 4 °C and washed twice with ice-cold PBS. Cell pellets were resuspended in 100 μl of polysome lysis buffer (100 mM KCl, 5 mM MgCl₂,10 mM HEPES pH 7.0, and 0.5% NP40) supplemented with 1 mM DL-Dithiothreitol, protease inhibitor cocktail (Sigma-Aldrich) and Ribonuclease inhibitor (Sigma-Aldrich), incubated on ice for 5 min and pelleted at $15,000 \times g$ for 15 min. The cleared supernatant was incubated with anti-RBM25-coated protein-A agarose beads (Sigma-Aldrich) for 4 h at 4 °C and then washed four times with ice-cold washing buffer (50 mM Tris-HCl, pH 7.4, 150 mM NaCl, 1 mM MgCl₂, and 0.05% NP40). The beads were resuspended in 100 μl washing buffer supplemented with proteinase K and incubated at 55 °C for 30 min to release the RNA-protein components. The RNA was isolated using Trizol (Invitrogen) and subsequently reverse-transcribed using the ProtoScript cDNA synthesis kit (New England Biolabs).

**RNA-Seq analysis**. RNA was extracted from $1 \times 10^6$ cells using the AllPrep DNA/RNA Mini kit (Qiagen) according to the manufacturer's recommendations. For RNA-seq library generation, 1 μg RNA was fragmented, size selected (350–500 bp),

reverse-transcribed according to the TruSeq RNA Library Prep Kit V2 sample preparation guide (Illumina). The libraries were analyzed by Qbit and Agilent Bioanalyzer High sensitivity assay (Agilent) and pooled in equimolar amounts. Multiplexed samples were sequenced on an Illumina HiSeq 2500 system at the Danish National High-Throughput DNA Sequencing Centre, University of Copenhagen (Copenhagen, Denmark). Each sample yielded 80–100 million 100-nucleotide paired-end reads.

For differential gene expression analysis, reads were mapped to human genome (hg19) using STAR (default parameters)[51]. Expression levels for GENCODE gene annotations (v19) were quantified using featureCounts[52]. Differential gene expression analysis between shRBM25 and shScr was performed using DESeq2[52] at an false discovery rate of <0.05.

Gene set enrichment analysis (GSEA; http://www.broadinstitute.org/gsea/index.jsp) was performed on shRBM25 vs. shScr triplicate expression files. For all gene sets, 1000 permutations and the Signal2Noise metric were used. Permutations by gene sets were conducted to assess statistical significance.

For the analysis of alternative splicing we used a computational pipeline involving the following steps: (a) Mapping of RNA-seq data to human genome (hg19) using TopHat v. 2.0.13[53], (b) Transcript quantification using Cufflinks v. 2.2.1[54]. (c) The resulting full-length transcripts were annotated with classes of alternative splicing using the Bioconductor package spliceR[26] with default settings.

**Additional bioinformatics analysis**. For TCGA AML data analysis, RNA-Seq data and survival data of AML patients of all cytogenetic risk groups were downloaded from the TCGA AML data portal[55]. BCL2L1-L expression was calculated as Log2 expression of the sum of Ensemble transcript IDs ENST00000307677 and ENST00000376062. BCL2L1-S expression was calculated as the expression of Ensemble transcript ID ENST00000376055. BIN1(+12) expression was calculated as the sum of the expression of exon 12-including Ensemble transcripts (ENST00000259238, ENST00000316724, ENST00000346226, ENST00000393040, ENST00000484253).

For the assessment of the correlation between *RBM25* levels and *MYC* target gene expression in patient samples, gene signatures from the MSigDB database were used to score patients from the TCGA cohort. Briefly, for each signature related to MYC, we computed the average of the log2-transformed gene expression values, using the expression of all genes in the signature normalized to the level of *MYC* expression. *MYC* normalization was performed in order to remove confounding differences in *MYC* expression levels. The TCGA patient cohort was then divided in two, based on the expression levels of *RBM25* in each patient, and gene signature scores compared between groups (Mann−Whitney *U* test).

**Statistics and general methods**. Sample sizes were not predetermined but are indicated in relevant figures. No blinding of experimental groups was performed. The number of times an experiment was performed is indicated in the figure legends. All statistical analyses were performed using Prism version 7.0 software (GraphPad Software Inc.). Comparison between groups was performed by a two-tailed, paired or unpaired Student's *t* test. The log-rank test was used to assess significant differences between survival curves. A nonparametric Mann–Whitney test was used for TCGA AML patient data analysis. Sample sizes chosen are indicated in the individual figure legends. *P* < 0.05 was considered to be significant. The results were represented as mean ± SD. *$P$ < 0.05, **$P$ < 0.01, ***$P$ < 0.001 and ****$P$ < 0.0001. Standard deviations are indicated on all figures.

## Data availability

Sequencing data have been deposited to the Gene Expression Omnibus (GEO) with the accession number GSE114027. Genome-wide transcriptome and mutational profiles of human AML data were obtained from The Cancer Genome Atlas data portal (http://www.cbioportal.org). The source data underlying Figs. 1a, c, 6a, b, and Supplementary Figs 1c, 1d, 4a are provided as Supplementary Files. All other data generated during this study are available from the authors on request.

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

## Acknowledgements

This study was supported through a center grant from the Novo Nordisk Foundation (Novo Nordisk Foundation Center for Stem Cell Biology, DanStem; Grant Number NNF17CC0027852) and a PhD fellowship from the Independent Research Fund Denmark to YG. The 8227 cells were a kind gift from Professor John E. Dick. We thank members of the Porse Lab for discussions.

## Author contributions

Y.G., N.H. and M.B.S. performed experiments. Y.G., M.B.S., S.P., N.R., N.S., F.O.B. and B.T.P. analyzed data. Y.G., M.B.S. and B.T.P. designed experiments. Y.G., M.B.S. and B.T.P. drafted the manuscript. All authors have proofread and approved the final version of the manuscript.

## Additional information

**Competing interests:** The authors declare no competing interests.

