## [Peer Review File · Nature Communications]

Reviewers' Comments:

Reviewer #1:

Remarks to the Author:

Ge et al. perform an in vivo pooled shRNA screen to identify splicing factors that promote or suppress tumor growth in a murine AML model. They validate their top hits in an in vivo competitive bone marrow transplantation assays using AML donor cells or normal hematopoietic cells. They identify the splicing factor RBM25 as a tumor suppressor in AML. Through a series of elegant functional assays, they demonstrated that RBM25 knock down (KD) promotes proliferation and decreases apoptosis of AML cell lines. Furthermore, they demonstrate that RBM25 controls splicing of BCL2L1 and BIN1 in AML cell lines, and that the splicing switching in BIN1 controls the activity of several downstream MYC target genes.

Findings from this study thus contribute to improving our understanding of splicing misregulation in tumorigenesis and highlight the role of non-mutated splicing regulators in tumor progression. In a very detailed work, the author provide a link between the levels of a splicing factor, RBM25, the splicing of its downstream target, BIN1, and the activation of MYC driven pathways. This study will be of great interest to the cancer and the RNA biology communities and should be published.

Several questions should be addressed prior to publication.

Major comments

1. The author state that they observe a “moderate correlation ($R=0.53$) between the biological replicates when comparing the fold change (FC) in shRNA representation between start and endpoint” (Page 6, lane 145-479). This raises the question of the reproducibility of this screen. Please clarify how reproducible are the top candidates, including RBM25. In addition, please explain why the top depleted hit, Nova (Table S3), was not selected for in vivo validations.
2. The author use a number of AML cell lines and normal hematopoietic cells, yet a description of the baseline protein and RNA levels of RBM25 is missing, making it difficult to interpret some of the results. For example, please show RBM25 protein levels after KD in Lp30, MLL-AF9 and c-kit++ control cells in Figure 2 and associated supplementary figures. Is RBM25 expressed at high levels in c-kit++ cells, and what are its levels after KD?
3. Overexpression (OE) of RBM25 in U937 cells does not increase the total RBM25 protein levels (Figure 3e), yet it decreases colony formation (Figure 3f). Please further discuss why this would be the case? Strikingly there is no difference between RBM25 levels in shSCR+RBM25OE vs. shRBM25+ RBM25OE (Figure 3e), and there is also no difference in their transforming capacity (figure 3f). Does exogenous RBM25 auto-regulate endogenous RBM25 levels through splicing, similarly to what was previously described for SRSF1 and other splicing factors? Also, please provide quantitative data for RBM25 levels in Figure 3e across multiple replicates.
4. The authors start the analysis by focusing on a specific subtype of AML which exhibits mutant CEBPA. Please clarify throughout the analysis what are the AML subtypes represented by each of the cell lines? For example are U937 and Kasumi-1 CEBPA mutant cell lines?
5. A detailed analysis of the characteristics of human tumors with high vs. low RBM25 levels is missing to fully appreciate the clinical impact on this study. What are characteristics of tumors that exhibit low RBM25 levels, and do any of these tumors exhibit mutations in splicing factors or epigenetic modifiers? How frequent are RBM25 high vs. low tumors? Are there any RBM25 mutations? Is there a link between the presence of CEBPA mutations and RBM25 levels? What AML subtypes exhibit a concomitant decrease in RBM25 levels and an activation of MYC targets? A critical information missing to understand the role of RBM25 in AML is a detailed analysis of its levels in the hematopoietic lineage during development. Are low RBM25 levels mostly found in progenitors cells?

6. The correlation between RBM25 levels and the activation of MYC target genes, or the levels of the BIN+12 isoform, although statistically significant, does not seem very impressive (Figure 8b,c). Since RBM25 regulates splicing of other RNA isoforms, this raises the possibility that other RBM25-regulated isoforms correlate better with RBM25 levels and survival in human AML tumors. For example, BCL2L1 isoform levels correlate better with RBM25 levels than BIN+12 levels (Figure 5d). The author performed RNA-seq but have not fully exploited this data. We suggest analyzing the RNA-seq data with a computational pipeline dedicated to splicing analysis (e.g. rMTAS or MISO) to derive a set of RBM25-splicing targets from the KD experiment, and then comparing it to TCGA or other publicly available tumor data to derive an RBM25 splicing signature and define its correlation with RBM25 levels and clinical outcomes in human tumors.

7. RBM5 KD was previously shown to be detrimental to growth of cancer cell lines as stated in the discussion. Yet here, the authors present evidence for an increase in tumor growth and colony formation. Please further discuss the differences, including in cell types, potentially underlying the tumor suppressive vs. tumor promoting roles of RBM25.

Minor comments

8. Please replace "splice factor" by "splicing factor" as this is the terminology used in the RNA biology and splicing field.

9. Please add references for the description of the spliceosome and splicing regulatory machinery in the introduction (page 3, lane 67-77)

10. Please add quantification to the CPSE assays (Figure 3).

11. The KD of the BIN+12 isoform only partially rescues the effect of RBM25 KD (Figure 7d). Please rephrase the sentence page 1 lane 355-358 to include this nuance.

12. Please add patient's number in Figure 8b and c, and show all data points as a dot plot.

Reviewer #2:

Remarks to the Author:

The manuscript by Ge et al. describes a series of functional-genetic studies aimed at exploring the role of RNA binding proteins and splicing regulators in AML. The study starts out with a technically elegant focused in-vivo RNAi screen in an experimentally tractable model of AML driven by mutant CEBPA. As one of the top hits, the authors find that RNAi-mediated suppression of RBM25, an essential splicing regulator, surprisingly accelerates the growth of CEBPA-mutant murine AML. Thorough validation studies (which also include cDNA rescue experiments and orthogonal CRISPRi-based experiments) confirm these effects in the primary screening model as well as in human AML cell lines. When investigating the mechanistic basis of these effects, Ge et al. find that perturbation of RBM25 promotes tumorigenesis in a dose-dependent manner, at least in part by regulating splicing of BCL-X and BIN1 transcripts. While RBM25 has been previously identified to regulate the equilibrium between the pro- and anti-apoptotic BCL-X transcripts BCL2L1-S and -L, the authors show that suppression RBM25 favors expression of the anti-apoptotic isoform BCL2L1-S and renders AML cells insensitive to ABT-263. Through systematic profiling of alternatively spliced transcripts, the authors reveal that suppression of RBM25 also favors expression of a specific isoform of the MYC inhibitor BIN1 that does not impede MYC's function. The functional relevance of this effect is demonstrated in elegant RNAi- and CRISPRi-based studies showing that suppression of RBM25 leads to an upregulation of MYC target genes. Based on these findings, the authors propose that the splicing factor RBM25 acts as an AML tumor suppressor by maintaining the

equilibrium of the pro- and anti-apoptotic BCL2L1 isoforms in favor of the anti-proliferative isoform, and by regulating MYC-inhibition via alternative splicing of BIN1 transcripts.

Overall, this a well-designed and well-executed study exploring the function of RNA binding proteins and splicing factors that have recently been implicated as important regulators in AML and other cancers. The technical quality of the primary in-vivo screen and subsequent validation and functional studies is high, and several key findings are confirmed using elegant rescue experiments and orthogonal methods. The key finding that RBM25 (an essential splicing regulator) exerts tumor-suppressive functions is an unexpected and very important finding that will be of great interest, not only to AML researchers but to the broader cancer research community. Therefore, I think this manuscript is highly suitable for publication in a major journal such as Nature Communications. Nevertheless, I have a few points that the authors may consider:

Major points:

1. The most intriguing (and important) finding of the study by Ge et al. is that a gene that (according to recent CRISPR/Cas9 screens) must be considered as generally essential for cell survival can exert tumor-suppressive functions in a dose-dependent manner. Although opposing functions have been described for some chromatin regulators in cancer, such effects (at least to my knowledge) have not been demonstrated for a core essential gene such as RBM25. While authors briefly touch upon this point in the discussion, this seems to be a major finding that should be highlighted and discussed more clearly (also in the abstract).

2. While some integrated analyses are presented in Supplemental Tables, it would be important to provide all primary screening data (i.e. raw read counts of all shRNAs in all replicates), as is common practice for NGS-based profiling studies. Figure 1c shows the performance of two RBM25 shRNAs, but it would be important to know how other RBM25 shRNAs contained in the library performed in the screen. Importantly, both technical challenges associated screens in an in-vivo setting, as well as the high dose-dependency that must be expected for RBM25, would explain that only some RBM25 shRNAs score – yet, it seems important to provide readers with the full picture.

3. While in-vivo RNAi screens in CEBPA-mutant leukemia seem technically sound, the presented in-vitro screen in c-Kit⁺ progenitors (Fig. 1c) seems much less convincing, most likely due to the relatively short screening time-frame of only 7 days, which together with the limited proliferative potential of primary c-Kit⁺ cells in culture seems problematic for detecting strong effects on proliferation and/or survival. Most importantly, while I understand the author's intention to decipher splicing regulators that are specifically required in AML but not in normal c-Kit⁺ cells, I wonder whether they would have excluded a tumor suppressive splicing regulator just because its knockdown also enhances proliferation of normal c-Kit⁺ cells. This would in fact be relevant for characterizing a putative tumor suppressor gene. Overall, I think the screen in c-Kit⁺ cells adds very little and should be de-emphasized.

Minor points

- 1) Line 101: "this variant (...) also targets a slightly..."
- 2) Line 159: cross-reference to Fig 1a and d, but instead of 1a -> 1b
- 3) Line 202: there is one dot too much before the cross-reference
- 4) Line 301: cross-reference to 5c, but 5d instead
- 5) Line 422: "...the latter via its role..."
- 6) Fig 1b: "High-throughput sequencing"
- 7) Fig 1d: the labelling of the YFP/GFP ratio plots with "shScr" and "shRbm25" at the Y axis is misleading, because it suggests that shScr is YFP-labelled in the upper panel and shRbm25 is YFP-labelled in the lower panel.
- 8) Figure legend of Fig1: Fig1f is not referred to in the legend, instead, Fig1e is referred to as upper and lower panel.

- 9) Fig 5b: indication of sample size is missing in the legend
- 10) Supplementary Fig 1b: the grey bar of "Mouse B" can hardly be seen
- 11) Supplementary Fig 4c: There is an arrow in the first FACS plot
- 12) Supplementary Table 2 and 3 legend: "1) multiple shRNAs (...) should occur in the in vitro..."

Point-to-point-response to the reviewer's comments:

Reviewer #1

Ge et al. perform an in vivo pooled shRNA screen to identify splicing factors that promote or suppress tumor growth in a murine AML model. They validate their top hits in an in vivo competitive bone marrow transplantation assays using AML donor cells or normal hematopoietic cells. They identify the splicing factor RBM25 as a tumor suppressor in AML. Through a series of elegant functional assays, they demonstrated that RBM25 knock down (KD) promotes proliferation and decreases apoptosis of AML cell lines. Furthermore, they demonstrate that RBM25 controls splicing of BCL2L1 and BIN1 in AML cell lines, and that the splicing switching in BIN1 controls the activity of several downstream MYC target genes.

Findings from this study thus contribute to improving our understanding of splicing misregulation in tumorigenesis and highlight the role of non-mutated splicing regulators in tumor progression. In a very detailed work, the author provide a link between the levels of a splicing factor, RBM25, the splicing of its downstream target, BIN1, and the activation of MYC driven pathways. This study will be of great interest to the cancer and the RNA biology communities and should be published.

Several questions should be addressed prior to publication.

Major comments

1. The author state that they observe a “moderate correlation ($R=0.53$) between the biological replicates when comparing the fold change (FC) in shRNA representation between start and endpoint” (Page 6, lane 145-479). This raises the question of the reproducibility of this screen. Please clarify how reproducible are the top candidates, including RBM25. In addition, please explain why the top depleted hit, Nova (Table S3), was not selected for in vivo validations.

Response:

We appreciate that the reviewer is raising the issue of reproducibility of our *in vivo* shRNA screen. We are fully aware that a major caveat of performing *in vivo* shRNA screens is limited coverage, which is a combined consequence of the fact that only a sub-fraction of the transduced cells possesses leukemia initiating cell (LIC) capacity and the limited number of cells it is possible to inject into a recipient. We determined the optimal shRNA pool size based on a preliminary *in vivo* screen, in which we varied the shRNA pool sizes used for transduction from 24 to 384 shRNAs/pool, carried out transductions and transplantation in duplicate and assessed correlation between the replicates. Not surprisingly, we found that the correlation decreased with increasing pool sizes (Reviewer Response Fig. 1a). Considering the need for an acceptable compromise between reproducibility and a feasible number of pools, we chose a pool size of 150 shRNAs for the final *in vivo* screen. This resulted in a correlation coefficient of 0.53, which was within the expected range according to the titration assay results. A note concerning assessment of the pool size has been inserted on page 6, end of second paragraph.

To further minimize the risk of false-positive hits and possible off-target effects of individual shRNAs, we set up the following criteria for candidate selection: Enriched or depleted candidates should have multiple shRNAs targeting the candidate genes, which were located in the 20 percentile of the most enriched/depleted shRNAs. Furthermore, none of these shRNAs should display a comparable enrichment or depletion in the *in vitro* screen performed on c-Kit⁺ cells. The targets picked for further validation all displayed good reproducibility between the two replicates (Reviewer Response Fig. 1b). Description of the picking criteria is unaltered from the original manuscript and is located on page 6, third paragraph.

As noted by the reviewer, the top depleted hit *Nova1* was included in the first round of validation. However, the finding that only one of the shRNAs targeting *Nova1* gave a decent KD (Reviewer Response Fig 1c), caused us to suspect that the observed phenotype was a result of off-target effects of the shRNA. We therefore decided to exclude this candidate for further *in vivo* validation. A note concerning the KD efficiency of the scoring shRNAs has been inserted on page 7, top.

Reviewer Response Fig. 1.

a *In vitro* titration of SF library size. Pool sizes range from 24 to 384 shRNAs. Scatter plot of Log₂-fold change of mouse A versus mouse B, and correlation between the two biological replicates versus different pool size. R-values represent Spearman correlation coefficients. **b** Reproducibility of the selected enriched and depleted targets. **c** The KD efficiency of depleted candidate *Nova1* by RT-qPCR.

2. The author use a number of AML cell lines and normal hematopoietic cells, yet a description of the baseline protein and RNA levels of RBM25 is missing, making it difficult to interpret some of the results. For example, please show RBM25 protein levels after KD in Lp30, MLL-AF9 and c-kit⁺⁺ control cells in Figure 2 and associated supplementary figures. Is RBM25 expressed at high levels in c-kit⁺⁺ cells, and what are its levels after KD?

Response:

To address this comment, we carried out Western blotting to assess the levels of RBM25 protein in Lp30, MLL-AF9 and cKit⁺ cells before and after KD with two individual shRNAs (Revised Supplementary Figure 2b). We observed effective KD in all cell subsets as well as a relatively lower basal level of RBM25 in cKit⁺ compared to the Lp30 and MLL-AF9 cells (Revised Supplementary Figure 2b). These findings are in good concordance with what we observe at the mRNA level (Revised Supplementary Figure 2c) and are described on page 8, fourth paragraph.

3. Overexpression (OE) of RBM25 in U937 cells does not increase the total RBM25 protein levels (Figure 3e), yet it decreases colony formation (Figure 3f). Please further discuss why this would be the case? Strikingly there is no difference between RBM25 levels in shSCR+RBM25OE vs. shRBM25+ RBM25OE (Figure 3e), and there is also no difference in their transforming capacity (figure 3f). Does exogenous RBM25 auto-regulate endogenous RBM25 levels through splicing, similarly to what was previously described for SRSF1 and other splicing factors? Also, please provide quantitative data for RBM25 levels in Figure 3e across multiple replicates.

Response:

We apologize that the WB shown in the original Fig 3e was not of sufficient quality to show a clear overexpression of RBM25 following transduction with an OE construct – this was largely caused by problems with an old batch of RBM25 antibody. We have now repeated the Western blot (in duplicates) using a new batch of RBM25 antibody, which strongly improved sensitivity (see Revised figure 3e). As expected, we see no difference in the colony numbers of the shSCR+RBM25OE and shRBM25+ RBM25OE samples as these express RBM25 at similar levels (see Revised figure 3f).

With respect to a potential RBM25 autoregulatory mechanism we have no data supporting this. Moreover, since we did not detect any changes in *RBM25* isoform usage following *RBM25* KD we consider such a mechanism quite unlikely.

4. The authors start the analysis by focusing on a specific subtype of AML, which exhibits mutant CEBPA. Please clarify throughout the analysis what are the AML subtypes represented by each of the cell lines? For example are U937 and Kasumi-1 CEBPA mutant cell lines?

Response:

We thank the reviewer for pointing out that our rationale behind using several AML-models in our validation efforts was not explained clearly enough. We have sought to use models from both mouse and human representing as many subtypes as possible in order to substantiate that the observed effects were conserved between species and in different leukemic contexts. The reason for working in two different murine models is explained on page 8, fourth paragraph. The mutational status of U937 is not well studied, which we have now emphasized on page 8, bottom in the revised manuscript. Kasumi-1 cells have the t(8;21) translocation, which is mentioned on page 9, end of second paragraph. Neither of the two cell lines are known to have any *CEBPA* mutations.

5. A detailed analysis of the characteristics of human tumors with high vs. low RBM25 levels is missing to fully appreciate the clinical impact on this study. **a)** What are characteristics of tumors that exhibit low RBM25 levels, and do any of these tumors exhibit mutations in splicing factors or epigenetic modifiers? **b)** How frequent are RBM25 high vs. low tumors? **c)** Are there any RBM25 mutations? **d)** Is there a link between the presence of CEBPA mutations and RBM25 levels? **e)** What AML subtypes exhibit a concomitant decrease in RBM25 levels and an activation of MYC targets? **f)** A critical information missing to understand the role of RBM25 in AML is a detailed analysis of its levels in the hematopoietic lineage during development. **g)** Are low RBM25 levels mostly found in progenitors cells?

Response:

We thank the reviewer for the comment and fully agree that a more detailed analysis of the human tumors will strengthen our study. To address the specific

questions, we have carried out the following analysis of the patient data already described in the manuscript:

Ad a)

To determine whether there is correlation between *RBM25* expression levels and mutational burden in AML patients, we scrutinized the TCGA cohort for total number of mutations and displayed the distribution of mutations in patient samples which were ranked based on *RBM25* expression (Revised Supplementary Figure 6a). This analysis revealed a direct correlation between *RBM25* expression and mutational load where we observed an increased number of mutations in the 10 percentile high-expressing *RBM25* samples. This is in good concordance with the notion that *RBM25* is a tumor suppressor. A high *RBM25* level might thus necessitate additional mutational events for a malignant phenotype to develop.

To investigate whether low *RBM25* levels are associated with particular AML subtypes, we carried out the aforementioned analysis on specific mutations belonging to the subclasses defined by Papaemmanuil et al. 2016³ in patients displaying the 10 percent lowest vs. highest levels of *RBM25* (Supplementary Figure 6a). As illustrated in (Revised Supplementary Figure 6a), this analysis revealed that the low-expressing patients displayed a complete absence of the APL-associated t(15;17) translocation - an observation that fits well with our demonstration that *RBM25* KD results in accelerated tumor progression and delayed myeloid differentiation. Conversely, the t(15;17) translocation is associated with good prognosis (Papaemmanuil et al., 2016) and a relatively differentiated phenotype. This observation is in good concordance with the observed mutual exclusiveness of low *RBM25* expression and the t(15;17) translocation. This analysis is described in the revised manuscript on page 13, bottom.

Ad b)

There is no standardized definition of high and low *RBM25* expression in AML. In order to assess the clinical relevance of our findings in the Lp30 murine AML model as well as *in vitro* in human cell lines, we performed a ranking of patients

from the TCGA dataset based on *RBM25* expression and found the correlation between expression and survival depicted in Figure 8a.

Ad c)

We have not been able to find any *RBM25* mutations in adult AML patients – neither by conducting literature searches or by querying publically available databases. However, in a recently published dataset for pediatric AML (Cerami et al., 2012), there are two cases out of 1025 studied samples with chromosomal deletions including the *RBM25* locus.

Ad d)

We found no correlation between *CEBPA* mutational status and *RBM25* expression. Specifically, patients with *CEBPA* mutations expressed *RBM25* mRNA levels similar to the remaining cohort (data not shown) and, concordantly, *CEBPA* mutations were neither over- nor underrepresented in patients with low *RBM25* expression (Revised Supplementary Figure 6a).

Ad e)

To address whether a concomitant decrease in *RBM25* expression and activation of MYC targets occurs associated with specific AML subtypes we partitioned the TCGA dataset into two distinct groups: Group 1 contains patients which display low expression of *RBM25* (low 25% percentile) and high level of MYC target gene expression (MYC target gene score > median) and Group 2 containing patients with opposite characteristics. We subsequently assessed which genetic lesions were differentially represented between them (Revised Supplementary Figure 6b). This analysis revealed that mutations in epigenetic modifiers and *NPM1* were overrepresented (albeit not significantly for the latter) in samples with low *RBM25* expression and high MYC score, whereas the opposite was the case for *t(15;17)* and *t(8;21)* translocations. The trend towards overrepresentation of *NPM1* mutations in samples with low *RBM25* expression and high MYC score is potentially interesting, since the *NPM1* gene has previously been shown to be a direct MYC target (Haggerty et al., 2003; Zeller et

al., 2003). It is tempting to speculate that an increased *trans*-activation potential of MYC resulting from low *RBM25* expression may promote increased NPM1 levels which may, in turn, provide a selective pressure for the acquisition of *NPM1* LOF mutations.

We have included the description above in the revised manuscript on page 14, third paragraph.

Ad f) and g)

To address this question, we assess the *RBM25* expression through murine and human myelopoiesis using the BloodSpot database (Bagger et al., 2016) (Revised Supplementary Figure 1j-k). As indicated, *RBM25* mRNA is fairly ubiquitously expressed from LT-HSCs and throughout myeloid differentiation. A remark about this has been made on page 7, last paragraph.,

6. The correlation between *RBM25* levels and the activation of MYC target genes, or the levels of the BIN+12 isoform, although statistically significant, does not seem very impressive (Figure 8b,c). Since *RBM25* regulates splicing of other RNA isoforms, this raises the possibility that other *RBM25*-regulated isoforms correlate better with *RBM25* levels and survival in human AML tumors. For example, *BCL2L1* isoform levels correlate better with *RBM25* levels than BIN+12 levels (Figure 5d). The author performed RNA-seq but have not fully exploited this data. We suggest analyzing the RNA-seq data with a computational pipeline dedicated to splicing analysis (e.g. rMTAS or MISO) to derive a set of *RBM25*-splicing targets from the KD experiment, and then comparing it to TCGA or other publicly available tumor data to derive an *RBM25* splicing signature and define its correlation with *RBM25* levels and clinical outcomes in human tumors.

Response:

We fully acknowledge that the correlation of *RBM25* levels with MYC target genes and BIN1(+12) levels in patient material is not overwhelmingly impressive. But since we are working on highly heterogeneous material where expression of downstream targets is influenced by a panoply of other factors,

this is to be expected. We are also aware that the effect of RBM25 levels on survival is driven by other downstream events than alternative splicing of BIN1 as well. However, we chose to focus on this particular mechanism, since it represents a previously uncharacterized modus of tumor growth regulation.

We thank the reviewer for the suggestion of further exploring the RNA-seq data and have earlier performed some initial analyses along these lines. Specifically, we identified the most de-regulated 28 isoforms (based on changes in isoform fraction) in the U937 knockdown experiments and compared these to splicing changes in the TCGA AML dataset between *RBM25* high and *RBM25* low patients. Somewhat surprising, only three isoforms (including BIN1(+12)) changed their levels in the same manner in U937 *RBM25* KD cells and AML patient material (*RBM25* low cells) thus precluding any meaningful signature analysis in this setting. There are a number of potential explanations for this observation: First, a substantial fraction of isoform changes observed following knockdown of *RBM25* in U937 cells could in principle be cell-type specific indirect effects. Second, and along the same lines, the heterogeneous nature of the TCGA dataset may require that we assess subtype specific changes in isoform usage, which we unfortunately don't have the power to do given the limited number of patients in this dataset. Thus with the present available data we're unable to do the proposed signature analysis in AML.

As mentioned in the discussion, we will soon embark on studies on *RBM25* in other human cancers (with more data), however this is beyond the scope of the present work.

7. *RBM5* KD was previously shown to be detrimental to growth of cancer cell lines as stated in the discussion. Yet here, the authors present evidence for an increase in tumor growth and colony formation. Please further discuss the differences, including in cell types, potentially underlying the tumor suppressive vs. tumor promoting roles of *RBM25*.

Response:

We thank the reviewer for the comment and agree that this question should be addressed more thoroughly. The phenotypic difference between CRISPR/CAS9 mediated complete KO in several human cancer cell lines (Carlson et al., 2017) and our down-modulation is indeed intriguing and we have now extended the discussion to further address this, page 16, bottom).

Minor comments

8. Please replace “splice factor” by “splicing factor” as this is the terminology used in the RNA biology and splicing field.

It is corrected.

9. Please add references for the description of the spliceosome and splicing regulatory machinery in the introduction (page 3, lane 67-77)

It is added.

10. Please add quantification to the CPSE assays (Figure 3).

It is added in Supplementary Figure 3a.

11. The KD of the BIN+12 isoform only partially rescues the effect of RBM25 KD (Figure 7d). Please rephrase the sentence page 1 lane 355-358 to include this nuance.

It is rephrased on page 12, top.

12. Please add patient’s number in Figure 8b and c, and show all data points as a dot plot.

It is added.

Reviewer #2

The manuscript by Ge et al. describes a series of functional-genetic studies aimed at exploring the role of RNA binding proteins and splicing regulators in AML. The study starts out with a technically elegant focused in-vivo RNAi screen in an

experimentally tractable model of AML driven by mutant CEBPA. As one of the top hits, the authors find that RNAi-mediated suppression of RBM25, an essential splicing regulator, surprisingly accelerates the growth of CEBPA-mutant murine AML. Thorough validation studies (which also include cDNA rescue experiments and orthogonal CRISPRi-based experiments) confirm these effects in the primary screening model as well as in human AML cell lines. When investigating the mechanistic basis of these effects, Ge et al. find that perturbation of RBM25 promotes tumorigenesis in a dose-dependent manner, at least in part by regulating splicing of BCL-X and BIN1 transcripts. While RBM25 has been previously identified to regulate the equilibrium between the pro- and anti-apoptotic BCL-X transcripts BCL2L1-S and -L, the authors show that suppression RBM25 favors expression of the anti-apoptotic isoform BCL2L1-S and renders AML cells insensitive to ABT-263. Through systematic profiling of alternatively spliced transcripts, the authors reveal that suppression of RBM25 also favors expression of a specific isoform of the MYC inhibitor BIN1 that does not impede MYC's function. The functional relevance of this effect is demonstrated in elegant RNAi- and CRISPRi-based studies showing that suppression of RBM25 leads to an upregulation of MYC target genes. Based on these findings, the authors propose that the splicing factor RBM25 acts as an AML tumor suppressor by maintaining the equilibrium of the pro- and anti-apoptotic BCL2L1 isoforms in favor of the anti-proliferative isoform, and by regulating MYC-inhibition via alternative splicing of BIN1 transcripts.

Overall, this a well-designed and well-executed study exploring the function of RNA binding proteins and splicing factors that have recently been implicated as important regulators in AML and other cancers. The technical quality of the primary in-vivo screen and subsequent validation and functional studies is high, and several key findings are confirmed using elegant rescue experiments and orthogonal methods. The key finding that RBM25 (an essential splicing regulator) exerts tumor-suppressive functions is an unexpected and very important finding that will be of great interest, not only to AML researchers but to the broader cancer research community. Therefore, I think this manuscript is highly suitable for publication in a major journal such as Nature

Communications. Nevertheless, I have a few points that the authors may consider:

Major points:

1. The most intriguing (and important) finding of the study by Ge et al. is that a gene that (according to recent CRISPR/Cas9 screens) must be considered as generally essential for cell survival can exert tumor-suppressive functions in a dose-dependent manner. Although opposing functions have been described for some chromatin regulators in cancer, such effects (at least to my knowledge) have not been demonstrated for a core essential gene such as RBM25. While authors briefly touch upon this point in the discussion, this seems to be a major finding that should be highlighted and discussed more clearly (also in the abstract).

Response:

We fully agree that this question should be addressed in more detail – and we believe that we have done so in our response to question 7 from Reviewer 1. Furthermore, we have now extended the discussion to this issue, page 16, bottom). Here we note that a similar behavior appears to hold true for the *SRSF2* and *SF3B1* splicing factors, both of which are essential for mouse development and are heterozygously mutated in MDS and AML.

2. While some integrated analyses are presented in Supplemental Tables, it would be important to provide all primary screening data (i.e. raw read counts of all shRNAs in all replicates), as is common practice for NGS-based profiling studies. Figure 1c shows the performance of two RBM25 shRNAs, but it would be important to know how other RBM25 shRNAs contained in the library performed in the screen. Importantly, both technical challenges associated screens in an in-vivo setting, as well as the high dose-dependency that must be expected for RBM25, would explain that only some RBM25 shRNAs score – yet, it seems important to provide readers with the full picture.

Response:

We thank the reviewer for emphasizing the importance of providing the raw screening data, which could be interesting to the other researchers in the field. We have therefore included the raw read data from both screens as separate sheets in Supplementary Data 1.

We have also tested the knockdown efficiencies of all *RBM25* targeting shRNAs in the library which demonstrates that the two scoring shRNAs in the Lp30 screen are also those with the best KD efficiency (See Revised Supplementary Figure 1h-i). We refer to this in the revised manuscript on page 7, last paragraph.

3. While in-vivo RNAi screens in CEBPA-mutant leukemia seem technically sound, the presented in-vitro screen in c-Kit⁺ progenitors (Fig. 1c) seems much less convincing, most likely due to the relatively short screening time-frame of only 7 days, which together with the limited proliferative potential of primary c-Kit⁺ cells in culture seems problematic for detecting strong effects on proliferation and/or survival. Most importantly, while I understand the author's intention to decipher splicing regulators that are specifically required in AML but not in normal c-Kit⁺ cells, I wonder whether they would have excluded a tumor suppressive splicing regulator just because its knockdown also enhances proliferation of normal c-Kit⁺ cells. This would in fact be relevant for characterizing a putative tumor suppressor gene. Overall, I think the screen in c-Kit⁺ cells adds very little and should be de-emphasized.

Response:

Regarding the *in vitro* screen in c-Kit⁺ cells, we fully agree that the screening time-frame is relative short but these cells only have limited self-renewal. In order to address the sensitivity of phenotype detection, we provide the growth curve of the c-Kit⁺ cells transduced with or without SF library during the selection (Reviewer Response Fig 2). As this figure illustrates, we were able to detect clear growth defect of the SF library-transduced cells, and a substantial reduction of a large number of the shRNAs (Fig. 1c lower panel). Since we mainly

aimed at identifying the AML specific targets, the criteria were set based on the different behavior between the AML and normal c-Kit⁺ screen.

Reviewer Response Fig. 2

Growth curve of c-Kit⁺ cells transduced with scramble shRNA or SF library shRNAs during the time frame of the screen.

We agree that targets displaying tumor suppressive capacity are potentially interesting whether or not they also restrict growth of normal c-Kit⁺ cells. We have therefore modified the originals tables of putative tumor suppressors (Supplementary table 1) and promoters (Supplementary table 2) to also include those that score in both leukemic and normal cells. We have then color-coded those that are specific to Lp30 AML screen. Although beyond the scope of the present work a number of common tumor suppressor hits are indeed interesting and could be targets for further studies

Minor points

1) Line 101: “this variant (...) also targets a slightly...”

It is corrected.

2) Line 159: cross-reference to Fig 1a and d, but instead of 1a -> 1b

It is corrected.

3) Line 202: there is one dot too much before the cross-reference

It is corrected.

4) Line 301: cross-reference to 5c, but 5d instead

It is corrected.

5) Line 422: "...the latter via its role..."

It is corrected.

6) Fig 1b: "High-throughput sequencing"

It is corrected.

7) Fig 1d: the labelling of the YFP/GFP ratio plots with "shScr" and "shRbm25" at the Y axis is misleading, because it suggests that shScr is YFP-labelled in the upper panel and shRbm25 is YFP-labelled in the lower panel.

It is corrected.

8) Figure legend of Fig1: Fig1f is not referred to in the legend, instead, Fig1e is referred to as upper and lower panel.

It is corrected.

9) Fig 5b: indication of sample size is missing in the legend

It is added.

10) Supplementary Fig 1b: the grey bar of "Mouse B" can hardly be seen

It is corrected.

11) Supplementary Fig 4c: There is an arrow in the first FACS plot

It is corrected.

12) Supplementary Table 2 and 3 legend: "1) multiple shRNAs (...) should occur in the in vitro..."

It is corrected.

References:

Bagger, F. O., Sasivarevic, D., Sohi, S. H., Laursen, L. G., Pundhir, S., Sonderby, C. K., Winther, O., Rapin, N., and Porse, B. T. (2016). BloodSpot: a database of gene

expression profiles and transcriptional programs for healthy and malignant haematopoiesis. *Nucleic Acids Res* *44*, D917-924.

Carlson, S. M., Soulette, C. M., Yang, Z., Elias, J. E., Brooks, A. N., and Gozani, O. (2017). RBM25 is a global splicing factor promoting inclusion of alternatively spliced exons and is itself regulated by lysine mono-methylation. *J Biol Chem* *292*, 13381-13390.

Cerami, E., Gao, J., Dogrusoz, U., Gross, B. E., Sumer, S. O., Aksoy, B. A., Jacobsen, A., Byrne, C. J., Heuer, M. L., Larsson, E., *et al.* (2012). The cBio cancer genomics portal: an open platform for exploring multidimensional cancer genomics data. *Cancer Discov* *2*, 401-404.

Haggerty, T. J., Zeller, K. I., Osthus, R. C., Wonsey, D. R., and Dang, C. V. (2003). A strategy for identifying transcription factor binding sites reveals two classes of genomic c-Myc target sites. *Proc Natl Acad Sci U S A* *100*, 5313-5318.

Papaemmanuil, E., Gerstung, M., Bullinger, L., Gaidzik, V. I., Paschka, P., Roberts, N. D., Potter, N. E., Heuser, M., Thol, F., Bolli, N., *et al.* (2016). Genomic Classification and Prognosis in Acute Myeloid Leukemia. *N Engl J Med* *374*, 2209-2221.

Vitting-Seerup, K., Porse, B. T., Sandelin, A., and Waage, J. (2014). spliceR: an R package for classification of alternative splicing and prediction of coding potential from RNA-seq data. *BMC Bioinformatics* *15*, 81.

Zeller, K. I., Jegga, A. G., Aronow, B. J., O'Donnell, K. A., and Dang, C. V. (2003). An integrated database of genes responsive to the Myc oncogenic transcription factor: identification of direct genomic targets. *Genome Biol* *4*, R69.

Reviewers' Comments:

Reviewer #1:

Remarks to the Author:

The authors have addressed our comments and the resulting improved manuscript is ready for publication.

Reviewer #2:

Remarks to the Author:

In their revised manuscript Ge et al. provide a substantial amount of additional experiments and analyses that in my view fully address comments raised by the reviewers. Specifically, in response to my questions the authors have (1) added important discussion points on essential functions of RBM25, (2) provided raw screening data and included additional analyses demonstrating a good correlation between KD efficacy and effect sizes (as another line of evidence that shRNA-mediated effects are on target); (3) provided additional information and analyses on the screen in Kit+ cells, its technical limitations and its utility for hit selection.

Overall, I am fully satisfied with the author's detailed response to my questions and concerns - their revisions have further corroborated key conclusions and strengthened the manuscript. I congratulate the authors to a very interesting manuscript – I have no further concerns or questions and fully recommend its publication.